# A highly reactive precursor in the iron sulfide system

Adriana Matamoros-Veloza [1,2], Oscar Cespedes [3], Benjamin R.G. Johnson[3], Tomasz M. Stawski[2,4], Umberto Terranova[5,6], Nora H. de Leeuw[5,6,7] & Liane G. Benning [2,4,8]

Iron sulfur (Fe–S) phases have been implicated in the emergence of life on early Earth due to their catalytic role in the synthesis of prebiotic molecules. Similarly, Fe–S phases are currently of high interest in the development of green catalysts and energy storage. Here we report the synthesis and structure of a nanoparticulate phase ($FeS_{nano}$) that is a necessary solid-phase precursor to the conventionally assumed initial precipitate in the iron sulfide system, mackinawite. The structure of $FeS_{nano}$ contains tetrahedral iron, which is compensated by monosulfide and polysulfide sulfur species. These together dramatically affect the stability and enhance the reactivity of $FeS_{nano}$.

[1] School of Mechanical Engineering, University of Leeds, Leeds LS2 9JT, UK. [2] School of Earth and Environment, University of Leeds, Leeds LS2 9JT, UK. [3] School of Physics and Astronomy, University of Leeds, Leeds LS2 9JT, UK. [4] German Research Centre for Geosciences, GFZ, 14473 Potsdam, Germany. [5] Department of Chemistry, University College London, London WC1H 0AJ, UK. [6] School of Chemistry, Cardiff University, Cardiff CF10 3AT, UK. [7] Department of Earth Sciences, Utrecht University, 3584 CC Utrecht, The Netherlands. [8] Department of Earth Sciences, Free University of Berlin, 12249 Berlin, Germany. Correspondence and requests for materials should be addressed to A.M.V. (email: A.MatamorosVeloza@leeds.ac.uk) or to L.G.B. (email: benning@gfz-potsdam.de)

The existence of solid precursors or intermediates prior to the formation of stable crystals is increasingly established, although many experimental and analytical challenges to characterize such entities remain[1]. Evidence for such highly reactive intermediate precursors, frequently structurally different from their bulk counterparts, is available for oxic systems; for example, metastable amorphous phases precede the formation of crystalline $CaCO_3$ polymorphs[2,3], while nanocrystalline phases are necessary precursors for gypsum formation in the $CaSO_4$ system[4,5], and Fe-oxo Keggin precede the formation of ferrihydrite in the Fe–OH system[6,7]. However, so far, in anoxic systems, and in the low-temperature Fe–S system in particular, only aqueous Fe–S clusters[8,9], and a conventionally assumed solid phase with a disordered and nanoparticulate mackinawite-like structure are known[8–13].

Despite increasing evidence of the existence of multiple solid precursors in the Fe–S system[1], the nature, structure and stability of early formed solid Fe–S precursors with structures different to that of mackinawite have so far not been documented. Such Fe–S phases have been hypothesized as potential membrane catalysts for the formation of prebiotic molecules and life's emergence on early Earth[14–16]. Furthermore, such reactive Fe–S phases are of prime interest for the potential green catalytic conversion of atmospheric $CO_2$[16,17], and for the development of sustainable, clean, and low-cost energy storage technologies[18,19].

With this work, we document the existence, identity and structure of a highly reactive nanocrystalline solid Fe–S precursor phase that is structurally different to mackinawite and that is a required precursor to the formation of mackinawite. This phase is, therefore, an important component of all further transformation reactions to more stable phases in the Fe–S system. For example, in anoxic low-temperature environments the formation of pyrite ($FeS_2$) proceeds via this phase through the mackinawite pathway (e.g., mackinawite → greigite ($Fe_3S_4$) [ ± → marcasite ($FeS_2$)] and pyrite ($FeS_2$)[9–11,20–22].

## Results

**Existence of FeS$_{nano}$ precursor.** Using two distinct and highly controlled chemical and fully anaerobic approaches (slow titration and a novel diffusion method; see Supplementary Methods) allowed us to quantify all stages in the nucleation, growth, stabilization, and transformations of solid phases in the Fe–S system from aqueous ions to crystalline mackinawite. We show that, contrary to previous studies[8–12], an additional solid-phase FeS precursor does exist. We have named this phase FeS$_{nano}$ and characterized it through multiple complementary high-resolution microscopic and spectroscopic techniques (Table 1).

Slowly titrating an aqueous $Fe^{2+}$ solution with NaHS, or diffusing $H_2S_{gas}$ into an aqueous $Fe^{2+}$ solution and allowing the pH to increase to just below 4.5, led to the formation of faint gray precipitates (Supplementary Figs. 1–3; Supplementary Tables 1, 2) that formed when less than 2% of the total initial ferrous iron was consumed (Supplementary Fig. 3a). We anaerobically separated, dried and analyzed these precipitates by X-ray diffraction (XRD) and the patterns revealed the presence of a phase with $d$-spacings of 12.1, 9.3, and 7.6 Å (Table 1 and Fig. 1a). Such $d$-spacings are entirely different from the Bragg peaks characterizing mackinawite (main $d$-spacing 5.0 Å; Fig. 1a arrow). Although lattice plane distances of up to 6.7 Å were reported for disordered mackinawite[12,23], the presence of three distinct $d$-spacings larger than 7 Å in our precipitate—from both the titration and diffusion experiments (Fig. 1a)—clearly documents the existence of a different, new solid Fe–S phase.

**FeS$_{nano}$ characteristics and transformation to mackinawite.** The relatively weak nature of these large $d$-spacing XRD peaks shows

### Table 1 Characteristics of the new phase FeS$_{nano}$ formed and stabilized at pH < 4.5

| Parameter | Characteristics | Technique |
|---|---|---|
| Size, nm | 2 | TEM |
| $d$-spacings, Å | 12.1, 9.3, 7.6 | XRD, TEM |
| S–S species | Monosulfides (62%), polysulfides (17%), disulfides (14%) | XPS, Raman |
| Fe–S species | $Fe^{II}$–S (~60%) and $Fe^{III}$–S (~40%) $Fe^{II}$ and $Fe^{III}$ | XPS XANES |
| Interatomic distances, Å | First shell $Fe^{II}$–S: 2.23, 4 S atoms | EXAFS |
| | Second shell $Fe^{II}$–Fe: 4.10, minimum 2 Fe atoms | |

that the new Fe–S phase is nanoparticulate, a characteristic also confirmed through high-resolution transmission electron microscopy (HR-TEM; Fig. 1b–e). However, a Pseudo-Voigt fitting procedure used on the XRD patterns allowed us to derive crystallite sizes, which suggested the coexistence of crystallites of ~23 and ~44 nm in size (Supplementary Fig. 4a and Supplementary Discussion). In contrast, the TEM images most often revealed highly monodispersed nanocrystals ~2 nm in diameter (Table 1 and Fig. 1b). The larger crystallite sizes derived from XRD peak fitting likely reflect the self-assembly or fusion of individual crystals to bigger agglomerates, as occasionally also observed by TEM (Supplementary Fig. 4b, c); furthermore, the hump at 10.8 $2\theta$ (8.2 Å) in the diffraction pattern from the diffusion reaction (Fig. 1a) results from short-range order in these larger structures. Although nanoparticle aggregates were also observed in TEM, during this sample preparation the samples were deposited onto the TEM grids from a dilute suspension. In contrast, upon drying the bulk samples for XRD analyses such aggregation was unavoidable.

Fast Fourier transform (FFT) of images of several individual nanoparticles from different replicate experiments corroborated the large $d$-spacings (~12, ~9, and 7 Å; Fig. 1d, e; Supplementary Table 3) from the XRD, indicating that the thus formed nanoparticles remained the same without further transformation. At the same time, energy dispersive analyses indicated a nonstoichiometric phase only containing iron and sulfur (Fig. 1f).

This new FeS$_{nano}$ phase could be stabilized by keeping the reacting solutions strictly anaerobic and at pH below 4.5. Although our data are from ambient temperatures, such conditions have been inferred as possible locus for Fe–S phase formation in microniches in acidic deep-sea hydrothermal vents[15]. When the pH was increased above 4.5 (Supplementary Fig. 3a), our new FeS$_{nano}$ transformed extremely rapidly (Supplementary Fig. 3a) to mackinawite, as evidenced by the replacement of the large $d$-spacings characteristic of FeS$_{nano}$ by Bragg peaks smaller than 6 Å, characteristic of mackinawite (e.g., Supplementary Fig. 3c; Supplementary Table 3).

The Raman spectra of FeS$_{nano}$ (Fig. 2a and Supplementary Fig. 5) showed two Fe–S asymmetric stretching vibrations at 204 and 215 cm$^{-1}$ and a symmetric Fe–S stretching vibration at 274 cm$^{-1}$ (Fig. 2a; Supplementary Table 4). The broadening of the peak at 204 cm$^{-1}$ indicates a high level of disorder at the local bond scale and a weakening in the interionic bonding caused by an expansion in the lattices[24].

Combining Raman with X-ray photoelectron spectroscopic (XPS) analyses of FeS$_{nano}$ also revealed the presence of variable proportions of long and short chain polysulfides (Fig. 2a, b, e; Supplementary Table 4) in its structure. The proportions of di- ($S_2^{2-}$) and polysulfides ($S_n^{2-}$) in the FeS$_{nano}$ phase contributed 14% and 17%, respectively to the integrated intensity signal, while

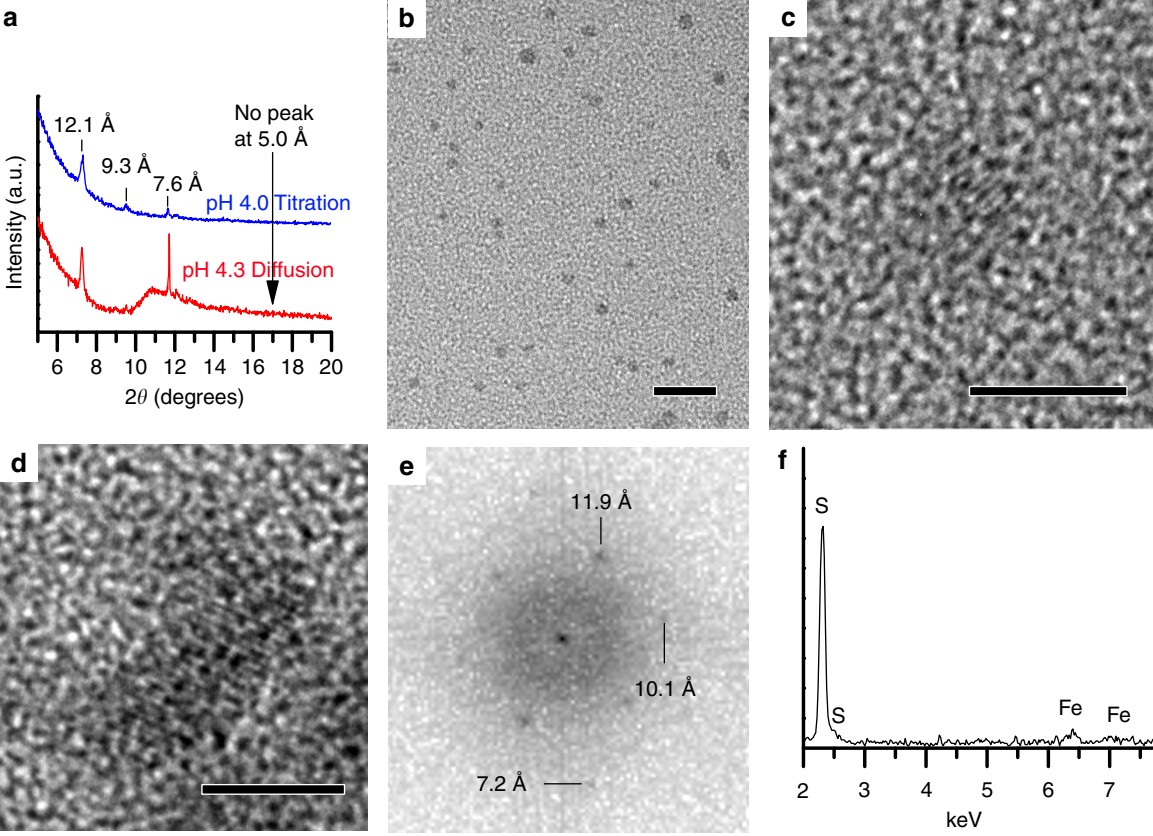

**Fig. 1** FeS$_{nano}$ formed at pH below 4.5. **a** XRD patterns of FeS$_{nano}$ synthesized under anaerobic conditions (a.u., arbitrary unit), with the patterns showing the three previously unknown low angle diffraction peaks; note absence of the characteristic Bragg peak for mackinawite at ~5.0 Å; the hump at 10.8 2$\theta$ (8.2 Å) suggests the presence of an agglomerated and poorly crystalline phase; **b** medium (scale bar 20 nm) and **c/d** high-resolution TEM images (scale bar 5 nm) of FeS$_{nano}$ nanoparticles formed in a diffusion and a titration experiment respectively; **e** FFT of the nanoparticle in **d** confirming the large $d$-spacings (Supplementary Table 3); and **f** EDX pattern confirming only Fe and S (1:8 ratio) in the FeS$_{nano}$ nanoparticles

the largest contribution (up to 62%) was from monosulfide species (S$^{2-}$) (Fig. 2b; Table 1 and Supplementary Table 6).

Polysulfides and Fe$^{III}$ species were previously documented in XPS analyses of crystalline Fe–S minerals (pyrite, pyrrhotite, and troillite), but in most cases their presence was attributed to surface oxidation during sample handling[25–29]. Similarly, polysulfides, Fe$^{II}$–S and Fe$^{III}$–S species found in mackinawite have been interpreted as a consequence of weathering[30].

In our case, the presence of polysulfides on the surface of the FeS$_{nano}$ phase were compensated by a 58:42 ratio in the Fe$^{II}$–S to Fe$^{III}$–S species (Fig. 2c, f; Table 1 and Supplementary Table 6) and they did not originate by surface oxidation (Fig. 2d, g). The presence of both Fe$^{III}$ and polysulfides in the FeS$_{nano}$ phase rather than only ferrous iron and monosulfides indicate defects in the initial atomic arrangements, which explain the highly reactive nature of FeS$_{nano}$, with the polysulfide pool providing a source of S for further redox reactions. Polysulfides are likely formed by electron transfer from monosulfides to the Fe$^{III}$ bonded in the FeS$_{nano}$ structure. Polysulfide species can be accommodated between the Fe and S arrangements of the FeS$_{nano}$ phase, as shown by density functional theory calculations (Supplementary Fig. 8 and text in SM), leading to the large $d$-spacings documented in the diffraction patterns and TEM images (Fig. 1). Such an expansion caused by the incorporation of similar molecular species has recently been evidenced in iron sulfide green rusts formed by reacting cysteamine with iron oxides[31]. That reaction reduced Fe$^{III}$ from iron oxide nanoparticles to Fe$^{II}$, yielding a Fe–S layered type material with large $d$-spacings that were the result of

the intercalation of cysteamine molecules in the green rust interlayers. Furthermore, it has even been suggested that precursors of mackinawite involve Fe$^{II}$Fe$^{III}$ hydroxide species related to the green rust group; however, no experimental work has demonstrated their existence and it is generally assumed that Fe$^{II}$ species lead the formation of the first condensed Fe–S phase[10].

**Local chemistry of Fe in FeS$_{nano}$.** The oxidation states and coordination of Fe in our synthetic FeS$_{nano}$ and in the transformation end-product mackinawite were derived from Fe K-edge X-ray absorption near-edge spectroscopy spectra (Supplementary Figs. 9, 10) and revealed clear differences in the preedge, edge jump and main peak (marked as regions I–III in Fig. 3a, b). The preedge in the FeS$_{nano}$ spectrum was 65% less intense than in mackinawite (i.e., total integrated area in FeS$_{nano}$ = 0.09 and in mackinawite = 0.26; Supplementary Table 9) indicating a different site symmetry. The shift in the centroid to higher energy indicates a slightly higher Fe$^{III}$ content in the FeS$_{nano}$. The lower intensity and higher energy of the edge jump in the FeS$_{nano}$ spectrum, compared to the mackinawite spectrum, are also consistent with the presence of Fe$^{III}$ in the FeS$_{nano}$ structure.

The local environment of Fe in our FeS$_{nano}$ phase, as analyzed by extended X-Ray absorption fine structure (EXAFS), revealed an average coordination of a first shell containing four S atoms in a tetrahedral coordination at a distance of 2.23 Å and a second shell containing at least two Fe atoms at 4.10 Å (Supplementary Figs. 11–13; Table 1 and Supplementary Table 11). There is a clear difference with mackinawite, in particular in the Fe–Fe bond

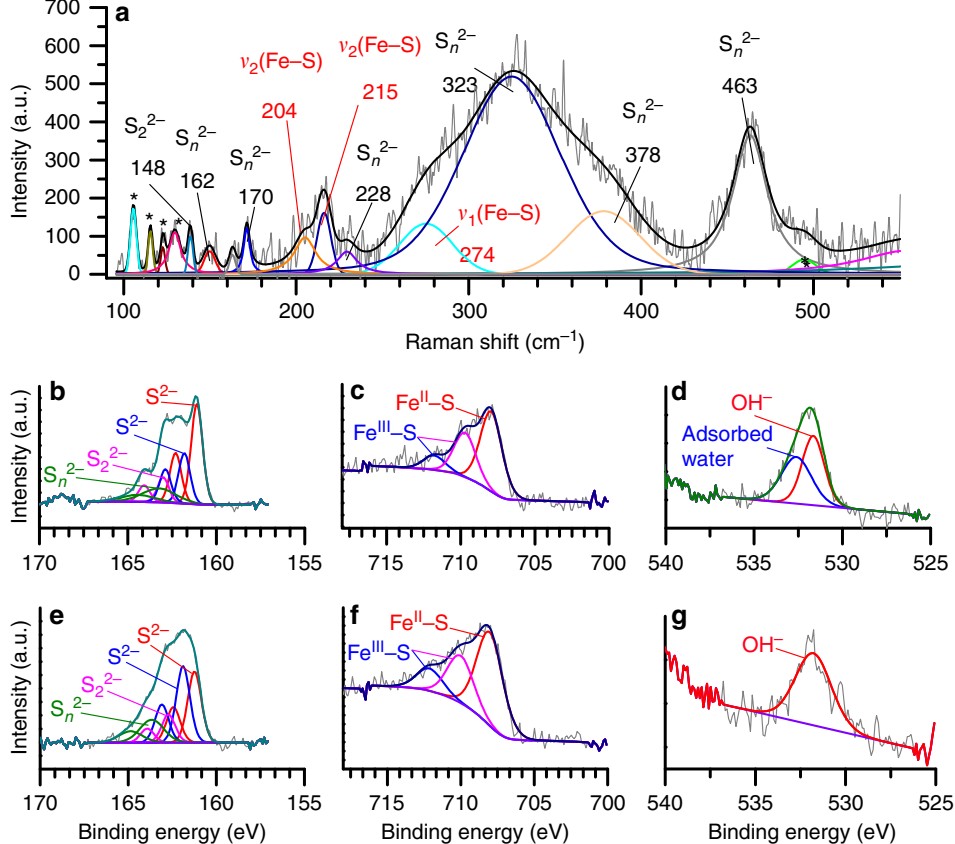

**Fig. 2** Raman and XPS data of $FeS_{nano}$. **a** Raman spectrum and deconvolutions of bands in $FeS_{nano}$ showing the asymmetric (204 and 215 cm$^{-1}$) and symmetric (274 cm$^{-1}$) Fe–S vibrations and various polysulfide species. Peaks marked with (*) are from the silicon grease from the sample holder. The asymmetric Fe–S vibrations at 204 and 215 cm$^{-1}$, with full width at half maximum (FWHM) of 15 and 9 cm$^{-1}$, respectively, correspond to monosulfides; **b**–**d** XPS spectra from $FeS_{nano}$ showing in **b** S2p with the doublet corresponding to the split of the spin-orbit into S2p$_{3/2}$ and S2p$_{1/2}$ in **c** Fe2p with fitting performed using the 2p$_{3/2}$ envelope and in **d** O1s high-resolution XPS spectra and fits for the $FeS_{nano}$ phase revealing the lack of Fe–O or S–O species; **e**–**g** corresponding to S2p, Fe2p, and O1s high-resolution XPS spectra from $FeS_{nano}$ after argon etching for five times 1 min. The XPS data indicate that sulfur atoms in $FeS_{nano}$ were present in two slightly different chemical environments (i.e., binding energies of 161.1 and 161.8 eV; Supplementary Table 6), that could reflect the presence of different atomic coordination arrangements[47]. The Fe and S species remained unchanged after etching; however, the relative contributions of the species after etching changed slightly, in particular for the S species (Supplementary Table 6). a.u. arbitrary unit

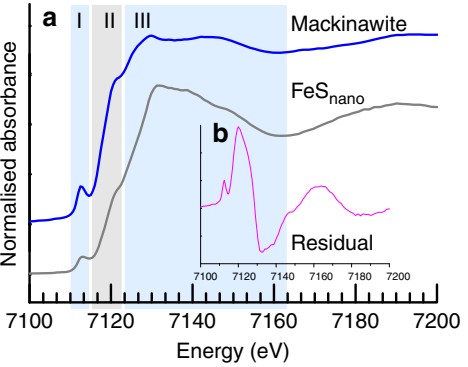

**Fig. 3** XANES data of $FeS_{nano}$ and mackinawite at early stages of formation. **a** Fe K-edge XANES spectra from $FeS_{nano}$ and mackinawite showing the (I) preedge, (II) increase in energy of the edge jump and (III) near-edge and **b** difference spectrum of mackinawite and $FeS_{nano}$ showing marked differences in the chemical environments between the two phases as revealed by the significant residual signal

distance (2.62 vs. 4.10 Å), indicating a higher degree of disorder, likely as a consequence of the inclusion of Fe$^{III}$ and polysulfides in its structure. The Fe–S bond distance derived in this study (2.23 Å) is similar to those reported in the literature (2.24 and 2.26 Å; [11]).

## Discussion

All combined, the above information allows us to derive the formation mechanism for the new $FeS_{nano}$ phase. In solution, the interaction between Fe$^{II}$ atoms and aqueous $H_2S$ at pH ≤ 4 leads to an atomic arrangement composed of one Fe atom and four S atoms in a tetrahedral coordination. This first atomic arrangement is the earliest stage in the nucleation of a solid and it follows on from the previously proposed aqueous clusters ($FeS_{aq}$ or $Fe_2S_2$)[13]. The acidic environment promotes the stabilization of both the aqueous Fe–S species[9,32,33] and their transition to the first structure solid Fe–S phase. We clearly demonstrate that this first solid in the Fe–S system is not mackinawite but a new phase, $FeS_{nano}$. With time, some of the Fe$^{II}$ from the initial Fe–S tetrahedra will lose electrons (oxidation to Fe$^{III}$). Such a partial "oxidation" leads to the formation of a layered solid with an unsaturated structure, with any excess of Fe$^{2+}$ being released back into the solution. The Fe$^{III}$ in turn will partially oxidize the aqueous $H_2S$ to form a series of polysulfide species to be accommodated between the Fe–S layers, forming the highly disordered $FeS_{nano}$ phase with large $d$-spacing and low Fe:S ratios (Table 1). This mechanism can be compared to the sulfidation of mackinawite, troillite and greigite in an $O_2$-free environment[34–36], yet in our case it describes the formation of a first solid phase from aqueous ions in the Fe–S system.

These results clearly document the pathways of formation and existence of a new solid $FeS_{nano}$ phase, which is the first solid

phase that forms in the Fe–S system and that is not a poorly ordered mackinawite but a new phase with totally different characteristics. This $FeS_{nano}$ phase consists of $Fe^{II}$ in tetrahedral and possibly $Fe^{III}$ in octahedral coordination, a structure that is balanced by the presence of polysulfides besides the dominant monosulfides. The new $FeS_{nano}$ phase can be stabilized for months if maintained in fully anaerobic conditions and at pH < 4.5. It is highly reactive and transforms to mackinawite even upon a small increase in pH, which makes it highly desirable as a potential catalyst material. Its existence opens new avenues in our understanding of the role of Fe–S nanophases in catalytic membranes, hypothesized to be important in the evolution of life on Earth, for which both low pH and high catalytic reactivity are crucial requirements[14,15]. The existence of metastable, transient, low pH $FeS_{nano}$ nanostructures may be relevant in microniches, for example in highly reduced deep-ocean hydrothermal vent systems. In such settings the open $FeS_{nano}$ nanostructures could have acted as catalysts in the development of prebiotic molecules and life's emergence on Earth. Finally, although as yet not tested, the ability to produce and stabilize such a highly reactive and nanocrystalline $FeS_{nano}$ phase at low temperatures will also undoubtedly find important applications in industrial catalysis, photovoltaics or electronics.

## Methods

**Titration experiments**. Ferrous sulfide phases were synthesized by the controlled and slow titration of a 150 mM NaHS solution into a 100 mM $Fe^{2+}$ solution. The starting solutions were made fresh at room temperature for each titration experiment using Mohr's salt [$(NH_4)_2Fe(SO_4)_2·6H_2O$] (ACS reagent Sigma Aldrich 99%), sodium sulfide [$Na_2S.9H_2O$] (Sigma Aldrich 99.999%) and deoxygenated 18.2 MΩ. cm milli-Q water[20,22]. All preparations were performed inside a glovebox (Coy-Laboratory Products Inc.), which was filled with a 5% $H_2$ and 95% $N_2$ gas mixture. The $O_2$-free conditions inside the glovebox were ensured through a Pd catalyst that reacted with the 5% $H_2$ to reduce any oxygen. The dry environment in the glovebox was maintained with silica gel. Full details of the preparation of $O_2$-free water is provided in the Supplementary Methods.

Once the starting solutions were prepared (100 mM $Fe^{2+}_{(aq)}$ and 150 mM $NaHS_{(aq)}$), they were used for the titration experiments that were carried out in a Labfors reactor (Infors HT, Switzerland—Supplementary Fig. 1). The reactor system consists of a double-jacketed glass reaction vessel and an overhead unit with port connections for pH–Eh electrodes, inlets and outlets connections for gas and liquids and a built-in stirrer. The addition of reagent solutions was controlled through an automated precision metering pump unit, and the controller recorded pH and Eh. Before the start of an experiment, the Labfors reactor was flushed for 1 h with $N_2$ (99,9995%) and the headspace of the reactant solutions was maintained anaerobic by a continuous $N_2$ flow warranting a slight positive pressure of the gas in the reactor. The $N_2$ gas outlet from the $N_2$ gas bottle was connected to the reactor inlet through an oxygen trap (LIOT-4, Agilent Technologies) to remove any remaining oxygen in the bottled $N_2$. Furthermore, a gas wash bottle containing a 6 M NaOH solution was connected in the gas outlet of the reactor to trap any excess $H_2S$ gas as an additional safety measure. To sample the solutions, a sampling system was attached to one of the reactor outlets via a T-connector. The system was assembled from flow-through, two-way valves linked to two syringes, one for flushing the system with $O_2$-free nitrogen and a second (also $N_2$ gas preflushed) syringe for removal of a sample (Supplementary Fig. 1). All experiments started by anaerobically transferring the light blue $Fe^{2+}$ solution into the reactor vessel; this initial $Fe^{2+}$ solution had a pH of 4.1. Once transfer was achieved and the overhead $N_2$ gas flow was set to be constant, the initial $Fe^{2+}$ solution was slowly titrated to pH 6–7 through the slow addition (0.47 mL/min) of the also anaerobic NaHS solution. The increase in pH from 4.1 to just below pH 7.0, required almost 9 h of continuous NaHS addition. The pH, Eh, volume of NaHS added, and time were recorded automatically every minute.

In the initial stages of the experimental work and to find optimal conditions in terms of time and pH ramp conditions, we carried out a series of test titration experiments with much higher concentrations of NaHS (1000, 500, and 150 mM; Supplementary Table 1). From these tests we determined that the optimal experimental parameters were slow titration with 150 mM NaHS. Thereafter, titration experiments with this concentration of NaHS were repeated three times to demonstrate that the data are highly reproducible in terms of time, pH ramp and added volume of NaHS. The continuously recorded pH profile allowed us to quantify exactly the volume of NaHS needed to reach certain pH conditions and thus to choose the most adequate sampling times. Samples were removed from the reactor at different pH values (defined by NaHS addition and time) through the double syringe sampling system. Once in the sample syringe, the samples were transferred immediately into an inflatable glove bag filled with oxygen-filtered $N_2$ gas for transfer back into the CoyLaboratory glovebox. Once in the glovebox,

samples were filtered using a vacuum filtration kit and 0.02 μm pore size polycarbonate membranes to retain the solids.

**Diffusion experiments**. All diffusion experiments were performed inside of the CoyLaboratory glovebox. Although the $Fe^{2+}$ solutions were prepared the same way as for the titration experiments (Mohr's salt [$(NH_4)_2Fe(SO_4)_2·6H_2O$]), the source of sulfide for the diffusion experiments was $H_2S$ gas that was generated by reacting HCl with solid $Na_2S·9H_2O$ crystals (Sigma Aldrich 99.999%). For each diffusion experiment, 25 mL of a freshly prepared $Fe^{2+}$ solution were transferred inside the glovebox into a 100 mL Schott Bottle (Duran®). In another Schott bottle, ~1 g of $Na_2S·9H_2O$ was mixed with 10 mL of 6 M HCl to generate $H_2S$ gas. This gas was allowed to diffuse and be transferred into the bottle containing the $Fe^{2+}$ solution through gas-tight tubing and thus increase the pH of the iron solution to ~4.3. The diffusion was allowed to proceed for 1 h. A 6 M NaOH solution was used as a trap to capture any excess of $H_2S$ in the reactor (Supplementary Fig. 2). Twelve diffusion experiments (Supplementary Table 2) were performed and the end pH of the mixed solutions was measured with a calibrated pH meter after 1 h of reaction. At the end of each diffusion experiment the solid precipitates were separated from the solution using a vacuum filtration kit and 0.02 μm pore size polycarbonate membranes. The filtrated solution was kept for total iron analysis by inductively coupled plasma optical emission spectrometry (ICP-OES; see below). The diffusion experiments were always performed fresh and the immediately separated solids were re-suspended on $O_2$-free ethanol, deposited on an analysis holder or substrate for analysis (see below) and left to dry inside the glovebox. These solid samples were analyzed with a variety of solid analyses methods, which each had special sample handling requirements (see details below and Supplementary Table 2). For each of these analyses, samples were removed from the glovebox on their respective substrates, but in sealed, double-jacketed containers that were placed inside of three layers of oxygen-free $N_2$ gas-filled and sealed bags.

**Solution characterization**. Total iron concentrations were analyzed in the supernatant solutions after separation of the solid phases after immediately diluting and acidifying with 1% $HNO_3$. Iron measurements were carried out by ICP-OES (Thermo Fisher iCAP 7400 Radial) with a calibration between 1 and 100 ppm. The quality of the measurements was controlled through duplicate analyses of the samples and analysis of QC check-standards before and after the analysis.

**Solid characterization methods**. For all solid analyses all sample handling and mounting steps were done inside the anaerobic chamber. Once mounted, all samples were sealed in special holders or containers for analyses or transport to analytical instruments.

**Powder XRD**. For XRD analysis, the as-synthesized anaerobic solid samples were filtered and redispersed in degassed ethanol and then mounted onto a flat silicon crystal inside an airtight Bruker XRD holder. This procedure was performed at all times in the anaerobic chamber under a strict and controlled oxygen-free environment. The airtight holder was placed inside an airtight container and was than inserted into another airtight container for transport to, and mounting into, the sample holder of the XRD instrument. The thus mounted samples were analyzed with a Bruker D8 diffractometer using a scan range between 2° and 70° $2\theta$ and at a scan rate of 0.05° $2\theta$/min. The XRD patterns were collected over 18–24 h in order to resolve even small and low angle peaks; during measurements changes in the background sometimes occurred (i.e., between 10° and 13° $2\theta$), yet the data is presented without further processing.

**High-resolution transmission electron microscopy**. Aliquots (either neat, diluted or redispersed in ethanol) of the solid samples were deposited onto holey carbon TEM grids (Agar Scientific), which were transferred into a special anaerobic transfer holder (Gatan 648 Double tilt). This allowed the safe anaerobic transfer of the highly reactive samples to the TEM instrumentation. All samples were freshly synthesized just immediately prior to TEM analyses. The effect of the dilution with ethanol was tested and compared to a nonethanol diluted sample and no changes were observed in the resulting images or spectral and diffraction information.

HR-TEM imaging, selected area electron diffraction and energy dispersive X-ray (EDX) spot analysis were acquired using a Field-Emission-Gun-Transmission-Electron microscope (FEI Tecnai TF20) fitted with a CCD Camera (Gatan Orius SC600A) and an EDX spectrometer (Oxford Instruments 80 mm² X-max). The microscope was operated at 200 kV. To avoid beam damage, the exposure for image collection was limited to maximum 0.25 s. Images were analyzed using the ImageJ software[37]. Atomic plane distances were calculated from the fast FFT analyses of several particles from the each acquired high-resolution image and from different areas in each image. The lattice planar distances were then calculated from the reciprocal values of the distance from the spots to the center (Supplementary Table 3).

**Raman spectroscopy**. The dried samples were transferred into a specially designed Raman analysis sample holder composed of two thin microscope cover slips, in between which the sample was sealed with silicone grease. The samples were freshly prepared and once in the sealed, airtight holder they were transferred from the

anaerobic chamber to the Raman instrument and stored until analysis in sealed airtight jars. Raman analyses were performed using a Horiba LabRam HR 800 microscope operated with lasers at excitations of 633 and 473 nm and using a 50× magnification objective, with gratings of 600 or 1800 ln/mm to give a resolution better than 1 cm$^{-1}$. The instrument was calibrated using a silicon standard (main Raman shift of 520.7 cm$^{-1}$) and Raman signals were acquired via a 1024 CCD detector. The instrument was operated in confocal mode using a 150–300 s detector exposure time and 20–30 spectral accumulations.

Several trials to determine the best combination of analysis parameters (e.g., laser wavelength, laser power, exposure time, number of accumulations, and background) were carried out prior to collection of the actual spectra; during these tests spectra were collected on different spots of various specially synthesized iron sulfide test samples. In addition, on a few samples a sequence of spectra was collected over time to evaluate any effects of the laser beam on the samples. From these tests the optimal conditions for analyses (473 nm laser using between 0.1% and 1% intensity ~1–10 mW) were determined. Using these parameters, we collected high-resolution spectra on all samples at wavelengths between 100 and 800 cm$^{-1}$. For each sample, several spots were analyzed by collecting 30 spectra with 150 s of exposure each. Once the spectral stack was collected, the data were evaluated to assure that no changes were induced during repeat analyses, after which the spectra were averaged.

All spectra were analyzed using the LabSpec 6 software and peaks were fitted using a combination of Lorentzian and Gaussian models. The goodness-of-fit was evaluated using the reduced $\chi^2$. The positions of the Raman vibrations were derived from the fits. Band positions and identities were assigned and compare with literature data (Supplementary Tables 4, 5).

**X-ray photoelectron spectroscopy.** Samples diluted in ethanol were deposited on gold substrates and then transferred to the XPS instrument inside double-walled plastic bags that were inside airtight jars. To mount the samples into the XPS instrument, a disposable glove bag was installed over the XPS sample interlock system. This glove bag was purged with $O_2$-free argon several times. The samples inside the airtight jar were transferred into the instrument, which was immediately evacuated to ultra-high vacuum (~10$^{-7}$ Pa) and samples were kept under vacuum overnight prior to analyses. During analyses the vacuum was maintained between 10$^{-7}$ and 10$^{-9}$ Pa.

XPS spectra were collected using a Thermo Escalab 250 XPS instrument equipped with a monochromatic AlK$_\alpha$ X-ray source (75–150 W). Calibration of the binding energies was performed using the carbon 1 s peak at 285 eV. Survey scans were collected between 200 and 800 eV with a pass energy of 160 eV. Depth of interaction was between 5 and 7 nm. The spot size was 500 μm and the analyses were done with a power of 150 W. High-resolution spectra were collected at the binding energies of Fe (700–740 eV), S (160–170 eV), O (525–545 eV) and C (280–295 eV), with a pass energy of 20 eV and a step size of 0.1 eV. The process was repeated after argon etching, 5 times each for 1 min, removing about 3–5 nm from the surface of the already analyzed samples.

The data were processed with the CasaXPS software and the background subtraction was done via the Shirley method, while fitting of the spectra was performed according to standard methods[38,39]. The high-resolution Fe2p spectra were analyzed for the 2p$_{3/2}$ envelope and fitted using a single peak for low spin Fe$^{II}$ species as in previous works[26,40]. (Supplementary Table 6). For Fe$^{III}$, multiplet splitting peaks were considered as predicted by the crystal field theory[41]. To build a model, several constrains were imposed (i.e., area of the peak, full width at half maximum (FWHM) and peak position) and the fitting did not rely on the combination of individual spectra as a fingerprint, but on the contribution of peaks with certain constrains that are characteristic for each specific species. The area under the curve (with a suitable baseline applied) was fitted with the minimum number of components to ensure best possible fit. The components were constrained so that the FWHM was maintained, whilst allowing free fitting of the peak area and position to ensure a good but scientifically viable fit. Where components were used to account for different bonding regimes, these were taken from the literature. Although this approach only provided initial values for peak positions, it still allowed for optimization of the peak areas through the fitting algorithm.

The FWHM was set at 1.8 eV for FeS$_{nano}$ and 1.0 eV for mackinawite. A satellite peak was included to fit the mackinawite spectrum. The S2p spectra were fitted using the 2p$_{1/2}$ and 2p$_{3/2}$ doublets, the first one was set to be half the area of the second (Supplementary Table 6). The fits were compared with data from the literature for different iron sulfides (Supplementary Tables 7 and 8). All spectra were corrected to allow for slight variations, using the C–C peak at 285 eV.

**X-ray absorption spectroscopy.** Fe K-edge X-ray absorption spectra (XAS) were collected at station I18 at the UK Diamond Light Source using a Si(111) mono-chromator with an energy resolution ($\Delta E/E$) of 1.4 × 10$^{-4}$. The monochromator was calibrated using a Fe-foil set at 7111.99 eV. Samples of the synthesized FeS$_{nano}$ and mackinawite phases were sealed inside the anaerobic chamber in between double sticky Kapton/mylar layers (with a total thickness of <200 μm) and transported anaerobically to the station in jars filled with $O_2$-free nitrogen. The particle size of the synthetic mackinawite used as a reference was ~5 nm as it was formed from the transformation of nanoparticulate FeS$_{nano}$ and it was aged only for short periods of time. The mackinawite spectra were collected between 6911 and 7580 eV, and the

FeS$_{nano}$ spectra between 6911 and 7400 eV. Both measurements were performed using a step size of ~0.5 eV. We also collected XAS transmission spectra from wustite, hematite and magnetite for reference.

The raw data were aligned, averaged, normalized and background-subtracted using the Athena software package[42]. The X-ray absorption near-edge structure (XANES) region of the Fe K-edge XAS spectra exhibit a characteristic preedge feature between 7109 and 7116 eV, composed of two overlapping peaks whose centroid depends on the contributions from Fe$^{II}$ and Fe$^{III}$ present[43,44]. Comparison with literature values revealed that our calibration was ~0.9 eV higher than Wilke et al. 2001[43].

Data from the EXAFS region were analyzed with the Artemis software package[42]. To test for Fe in tetrahedral and octahedral coordination in the FeS$_{nano}$ phase, theoretical models based on the structures of greigite and mackinawite were generated using the FEFF6 and ATOMS software packages[45]. Single scattering paths were considered in the fits (Fe–S, first shell and Fe–Fe, second shell) based on the greigite structure. However, only the Fe–S path could be considered for the mackinawite data as there was no contribution of the Fe–Fe path to the data. In the fits, the amplitude parameter ($S_0^2$) was fixed to 0.80 for both paths, which was derived from the synthetic mackinawite standard. The energy shift ($\Delta E_0$) was constrained to be the same for the two paths and it was set as a fitting parameter, as well as the disorder parameter in the distribution of interatomic distances ($\sigma^2$). Due to the small size of the particles, the coordination numbers ($N$) are not expected to be round figures and thus the coordination numbers could also be set as fitting parameters. To construct the fits, independent variables ($N_{var}$) were at least half of the independent points ($N_{ind}$). The data were then fitted through a least squares approach in the k-space between 2.8 and 6.5 Å$^{-1}$ using multiple k-weights. The quality of the fits was assessed with the reduced $\chi^2$ and the R-factor[46]. Coordination numbers ($N$), bond distances ($R$), Debye–Waller factors ($\sigma^2$) and inner potential corrections ($\Delta E$) were extracted from the fits.

**Data availability.** The authors declare that data supporting the findings of this study are available within the paper and in the supplementary information file or are available from the corresponding author upon reasonable request.

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

## Acknowledgments

We acknowledge the Natural Environment Research Council (Grant NE/J008745/1) for financial support for this work. We thank F. Mosselmans for helpful comments and discussion about the XAS data processing and interpretation. We acknowledge A. Kroner from Diamond Light Source Ltd. (UK) for beamtime allocation through industrial access. We also thank M. Ward and the Leeds Electron Microscopy and Spectroscopy Centre for help with TEM imaging and analyses. We thank R. Raiswell, M. Wolthers, and M. Krom for their insightful comments. L.G.B. also acknowledges financial support from the Helmholtz Recruiting Initiative. This work used the ARCHER UK National Super-computing Service (http://www.archer.ac.uk) and the UCL Legion High Performance Computing Facility (Legion@UCL) with its associated support services.

## Author contributions

A.M.-V. and L.G.B. conceived the study. A.M.-V. acquired and analyzed the data. O.C, B. R.G.J., and U.T. helped to conduct some experiments. A.M.-V., O.C., B.R.G.J., T.M.S., U. T., N.H.de.L., and L.G.B. interpreted the data. U.T and N.H.de.L. performed the density functional theory simulations. A.M.-V. wrote draft and revised manuscript. A.M.-V. and L.G.B. revised the scientific content. A.M.-V., L.G.B., O.C., B.R.G.J., T.M.S., U.T., and N. H.de.L. approved the final version of the manuscript.

## Additional information

**Competing interests:** The authors declare no competing interests.

