## [Peer Review File · Nature Communications]

Reviewers' comments:

Reviewer #1 (Remarks to the Author):

The authors report on the occurrence of a nano-crystalline FeS phase on the reaction-pathway of machinawite synthesis. This system is investigated in the context of role of minerals during the formation of pre-biotic molecules. Particularly, the high surface to volume ratio of nano-sized particles offers many chances for catalytic activity as exploited in catalytic reactions in the field of synthetic chemistry.

The article is well written and result are well documented although I find it quite difficult to find important details of analysis as they are spread over main text and supplementary. I guess, this is unavoidable in this paper format. The authors can well document that the new phase is different from mackinawite. The strongest constraints come from TEM and XRD. All further characterizations by XPS and XANES/EXAFS I do find less convincing, because I consider some of the data as over-interpreted (see below). In the current state of the analysis, they only document that this phase is different to mackinawite, all other assignments are not well constrained or are based on an analysis that is wrong or at least not constrained (XANES pre-edge).

In conclusion, this study reports on a very interesting and important observation, which certainly provides new insights to elucidating the role of minerals in the formation or conversion of organic macro-molecules. However, the authors are stretching the limits of some of their analytical techniques and should be more honest on the significance of their data. Even though controversial contributions are likely welcome in this journal one should stick to this principle. Therefore, I recommend publication only after these major revisions have been done.

Specific comments:

XRD and TEM provide very good constraints for formation of a phase that is different from mackinawite. The XRD pattern for the pH 4.3 samples shows additional signal from non-crystalline matter between 10 and 12 degr. 2-theta. Is there other indication for non-crystalline phases? I could not find a note by the authors on this feature in the XRD, which represents some clear difference between the two methods of synthesis. Please, discuss this.

XANES/EXAFS: The authors present XANES data on the sample and a mackinawite reference sample. There is a significant difference between the two sample constraining the assigned difference in crystal structure. In the analysis, the authors use a scheme developed on oxides for the interpretation of the pre-edge. Whether this works for sulfides has not been shown to my knowledge. In addition, the way of analyzing the pre-edge peak with two peaks (one for Fe²⁺ and one for Fe³⁺) has not been used at all in the literature and is likely not valid. In oxides, the peak for sixfold or fourfold coordinated Fe²⁺ is a doublet by itself. Same holds for octahedral Fe³⁺. Thus, the assigned Fe²⁺ and Fe³⁺ contributions for the FeS-sample are completely unconstrained by the XANES data. Maybe, the slight shift to higher E provides a qualitative hint. The pre-edge is related to a 1s-3d transition and thus non-dipolar in nature. i.e. for a centrosymmetric site it is forbidden and only possible through quadrupolar mechanism (low intensity). It becomes dipole allowed by p-d mixing, e.g. by loss of centro-symmetry. In essence, without presenting further sulfide model compounds and/or simulated spectra, the authors can only document a difference in site symmetry for Fe between FeS and mackinawite, with FeS having with a more centro-symmetric site than mackinawite. E.g. this could mean conversion from tetrahedral to octahedral symmetry or to square-planar symmetry. The authors need to rewrite this part and they should include more XANES data on sulfides at least from the literature (e.g. Prietzel et al. 2007 European Journal of Soil Science, 58, 1027). Finally, the

assignments to areas I, II, III with 1s-3d, 1s-4s, 1s-4p is not referenced at all, and only the first one is likely correct. 1s-4s transitions are also dipole-forbidden, so I doubt that it could be that strong. 1s-4p might be ok, but these states are already quite delocalized. In the absence of a DOS calculation, I would not quote any orbital assignment here. For the sake of the paper the terms I, II, III are fine. Please, do so.

The authors claim a difference in the Fe-O distance: 2.23 vs. 2.24 Å. Is this a significant difference given the limited EXAFS range for the FeS? I doubt that. Again, there is a difference to mackinawite, but only in the second shell. The mackinawite spectrum itself has no indication for signal by a second shell at all (Sup. Fig12 a +b), particularly not at 2.6 Å. In contrast, FeS shows some contribution by a second shell, as documented. So, I cannot follow the arguments of the authors from these data. There is no indication for Fe³⁺ in these EXAFS data, only some Fe-Fe contribution. Just from the Fourier Transform of the two spectra I would consider mackinawite to be more disordered than FeS, because it really shows no second-shell contribution. Please, rewrite accordingly or present stronger constraints.

Line 95: Anaerobic XRD? Please, write "XRD patterns of FeS synthesized at anaerobic conditions"

Line 101: What is the Fe/S ratio based on? Simple intensity ratio or by using a matrix correction for analyzing the EDX pattern?

Figure 2b-g: What are the peak assignments based on? Are the peak positions constant no matter what the compound is? Especially, in case of the S-spectra the choice looks arbitrary to me with the limited info provided.

Line 363: Why only crystal-field splitting for Fe³⁺? Again, comparing the fit in the supplement on mackinawite with the one on FeS, gives the impression that the fit model are chosen arbitrarily, at least for a non-XPS reader. I would rather see a direct comparison of spectra including spectra of other model compounds, at least for other members of the FeS-family.

Line 403: why between 3 and 6 Å⁻¹. The EXAFS spectra are much longer. Is this a typo? If not the authors need to explain this in more detail.

Reviewer #2 (Remarks to the Author):

The precipitation mechanism of Fe-S phases is important for a range of reasons as the authors indicate. The study uses a multi-modal analytical approach to determine the composition of the initial precipitate formed in the titration of a Fe(II) solution with an H₂S solution or a Fe(II) solution exposed to H₂S gas. The unexpected result is that a nano phase is formed which appears to contain both appreciable amounts of Fe(III) and polysulfides (i.e., S with an average oxidation state in excess of -II). This is unexpected because the phase this precursor leads to is mackinawite which is a slightly iron-rich FeS monosulfide phase, with all Fe(II) and S(-II). Hence, the results presented here indicate an initial precursor phase that is on balance more oxidized than the final product. The simplest interpretation of the data would be to assume that there is initially some minor amount of O₂ or other oxidant present that rapidly reacts and leads to the precursor, which then converts to mackinawite. The entire interpretation depends on how well oxygen or other oxidants are avoided in the experimental setup and how well one can avoid O₂-driven alteration during any of the analyses. The authors clearly realize this and in the supplemental material a considerable amount of attention is given to this issue.

However, this reviewer is concerned that the authors need to provide more details or present several control experiments to truly convince readers that the conclusions are based on experiments in which molecular oxygen or other oxidants did not play a role.

For example, it is posed in line 217 that deoxygenated water is used. There is NO indication how the water was deoxygenated, nor was a measurement of Dissolved Oxygen reported of this water. Placing DI with some slight amount of DO in an aerobic glove box does not ensure that the solution will be devoid of O₂. Sparging is probably the best method to remove DO, but earlier work by Butler et al shows that even with prolonged sparging some DO may remain, although the source of N₂ in this study may have a lower O₂ content because of the careful treatment with a trap. In any event, a control experiment to demonstrate the level of remaining O₂ is necessary. Furthermore, more recent work has shown that one has to also worry about UV irradiated water containing traces of H₂O₂. The study reports that Milli-Q water is used. Typically these units also include a UV lamp which leads to the formation of H₂O₂. Our experience is that one has to keep the water in the dark for a month or so to dissipate all H₂O₂ (Cohn, et al, Comparison of fluorescence-based techniques for the quantification of particle-induced hydroxyl radicals. Particle and Fibre Toxicology, 2008. 5.). It might be that the water used here was not irradiated, but those details should be spelled out.

Perhaps a greater concern is possible alteration of samples during analysis. It is not clear how one would go about doing meaningful control experiments to reject the simplest interpretation that there is some initial oxidation. Other studies on this system have used in situ synchrotron-based pair distribution function analysis and concluded that 'amorphous FeS' had the mackinawite structure (Michel, F.M., et al., Short- to medium-range atomic order and crystallite size of the initial FeS precipitate from pair distribution function analysis. Chemistry of Materials, 2005. 17(25): p. 6246-6255). Because the authors are trying to evaluate a system with only 2% of the iron precipitated, the PDF technique would not be useful as an in situ technique as the signal is likely way too low. However, it might be useful to think how one might as much as possible conduct an in situ experiment. For example, would it be possible to use a Raman fiber optic probe to study the material in situ. Any evidence that would not rely on sample pretreatment followed by ex situ analysis would strongly lend support for their conclusion.

Finally, I think it is important that the author present a more thorough analysis of the implications of the change in oxidation state of Fe and S in their system. If there is really nearly 40 % Fe(III), what are the energetics of this reverting back to Fe(II) to produce mackinawite.

With regard to the implications, the argument that this may be important on the early Earth is an overstatement. Even with a high CO₂ concentration in the atmosphere, the pH of the ocean would have been around 5 at best and the precipitation is likely to occur where highly alkaline solution flow into this ocean water. So the experiments here actually suggest that this precursor only plays a role below pH 4.1.

I think this is a well-conceived study, but given the implications more evidence that there was not some oxidant present either during the experiment or beam/irradiation induced alteration of the precursor is needed.

Martin Schoonen

Response to Reviewer #1 comments:

The authors report on the occurrence of a nano-crystalline FeS phase on the reaction-pathway of mackinawite synthesis. This system is investigated in the context of role of minerals during the formation of pre-biotic molecules. Particularly, the high surface to volume ratio of nano-sized particles offers many chances for catalytic activity as exploited in catalytic reactions in the field of synthetic chemistry.

The article is well written and result are well documented although I find it quite difficult to find important details of analysis as they are spread over main text and supplementary. I guess, this is unavoidable in this paper format. The authors can well document that the new phase is different from mackinawite. The strongest constraints come from TEM and XRD. All further characterizations by XPS and XANES/EXAFS I do find less convincing, because I consider some of the data as over-interpreted (see below). In the current state of the analysis, they only document that this phase is different to mackinawite, all other assignments are not well constrained or are based on an analysis that is wrong or at least not constrained (XANES pre-edge).

In conclusion, this study reports on a very interesting and important observation, which certainly provides new insights to elucidating the role of minerals in the formation or conversion of organic macro-molecules. However, the authors are stretching the limits of some of their analytical techniques and should be more honest on the significance of their data. Even though controversial contributions are likely welcome in this journal one should stick to this principle. Therefore, I recommend publication only after these major revisions have been done.

Reply: We thank reviewer 1 for the positive comments about our TEM and XRD data and the fact that indeed that data shows that our new FeS_{nano} phase is different than mackinawite; we address the concerns regarding data interpretation of Raman, XPS and XAS below.

Specific comments:

R1 Comment 1: XRD and TEM provide very good constraints for formation of a phase that is different from mackinawite. The XRD pattern for the pH 4.3 samples shows additional signal from non-crystalline matter between 10 and 12 degr 2-theta. Is there other indication for non-crystalline phases? I could not find a note by the authors on this feature in the XRD, which represents some clear difference between the two methods of synthesis. Please, discuss this.

R1 Comment 1 reply: We have not been clear enough about this and have now explained that the additional broad band between 10 and 12 degree 2-theta, which corresponds to 8.2 Å I due to the poorly crystalline nature of our phase.

To clarify this point, we modified the text in revised manuscript line 75: “...furthermore, the hump at $10.8\ 2\theta$ ($8.2\ \text{\AA}$) in the diffraction...”; we have also highlighted this in the Figure caption of Fig 1 (line numbers 96,97): “...the hump at $10.8\ 2\theta$ ($8.2\ \text{\AA}$) suggests the presence of a agglomerated and poorly crystalline phase; ...”

R1 Comment 2: XANES/EXAFS: The authors present XANES data on the sample and a mackinawite reference sample. There is a significant difference between the two sample constraining the assigned difference in crystal structure. In the analysis, the authors use a scheme developed on oxides for the interpretation of the pre-edge. Whether this works for sulfides has not been shown to my knowledge. In addition, the way of analyzing the pre-edge peak with two peaks (one for Fe^{2+} and one for Fe^{3+}) has not been used at all in the literature and is likely not valid. In oxides, the peak for sixfold or fourfold coordinated Fe^{2+} is a doublet by itself. Same holds for octahedral Fe^{3+} . Thus, the assigned Fe^{2+} and Fe^{3+} contributions for the FeS-sample are completely unconstrained by the XANES data.

R1 Comment 2 reply: We indeed had not detailed that the approach we used was certainly developed for oxides but has since successfully also been applied to silicate glasses, hydrous melts, and important for our purposes - sulfides and oxide-sulfide mixtures (Wilke et al. 2001 American Mineralogist, 86: 714-730; Berry et al 2003 American Mineralogist, 88, 967-977; Humphreys et al 2015 Journal of Petrology, 56(4): 795-814; Prietzel et al 2007, European Journal of Soil Science, 58, 1027-1041-reference recommended by the reviewer 1).

In particular Prietzel et al (2007) demonstrated that the centroid of the pre-edge (which is related to the positions of the individual components) is reasonably accurate to estimate the $\text{Fe}^{2+}/\text{Fe}^{3+}$ ratio of an unknown sample composed by sulfides and oxides comparing the centroid position to those of the standards. Therefore, there is evidence that this approach has been applied to sulfides, although Prietzel et al did not elucidate the peak components (deconvolution of peaks). However, see also reply to comment 4 below referring to shifts in Prietzel et al. (2007)

We now clarify this point in line numbers 155-165 of the revised Supplementary Material:

“To fit the XANES data, we used the method developed for oxides and later applied to silicate glasses, hydrous melts, sulfides and oxide-sulfide mixtures (9, 10, 14, 15). These studies have demonstrated that the centroid of the pre-edge (which is related to the positions of the individual components) is a reasonably accurate to estimate the $\text{Fe}^{\text{II}}/\text{Fe}^{\text{III}}$ ratio of an unknown sample composed of sulfides and oxides, when comparing the centroid position to those of the standards (15). Moreover, the peak positions of the deconvoluted peaks correlate with Fe^{II} and Fe^{III} components, with the average position corresponding to the pre-edge position of the centroid (9, 10). The approach of using the centroid of the peak has been well documented for sulfides, and the deconvolution of the peak has been demonstrated to be

accurate to estimate the FeII and FeIII ratio. Thus, we used this method to identify the presence and determine the proportions of these two oxidation states of Fe in the FeSnano."

In response to the reviewers comment that: "In addition, the way of analyzing the pre-edge peak with two peaks (one for Fe²⁺ and one for Fe³⁺) has not been used at all in the literature and is likely not valid" we can only state that maybe we do not fully understand this point but we do not agree that the use of two peaks for pre-edge fitting has not been used at all in the literature. We followed the procedures developed by Wilke et al., 2001 and later applied by Berry et al 2003 and Humphreys et al 2015. They fitted the pre-edge of several Fe compounds to two peaks, although during fitting they allowed the use of up to three peaks. These studies also established that the peak positions of the deconvoluted peaks correlate with Fe²⁺ and Fe³⁺ components, with the average position corresponding to the pre-edge position of the centroid. The approach of using the centroid of the peak has been well documented for sulfides, and the deconvolution of the peak has been demonstrated to be accurate to estimate the Fe²⁺ and Fe³⁺ ratio. Thus, we used this method to identify the presence and proportions of these two oxidation states of Fe in the FeSnano. Moreover, we showed that the energy positions of the two deconvoluted peaks agree with these previous studies. So we are not sure why the reviewer states that this approach is erroneous.

To fully explain that the use of 2-3 peaks for fitting XANES data has been established in the literature, we added in line numbers 170-171 of the revised supplementary material the following:

"...and in agreement with pre-edge fitting to two or three peaks reported in literature (9,10)."

We appreciate the reviewer's comment about the fact that different coordinations of Fe²⁺ and Fe³⁺ should exhibit peak components for each electronic transition, according to the crystal-field theory (i.e., four transitions for Fe²⁺ in square planar or tetrahedral coordination; three transitions octahedral coordinated Fe²⁺; two transitions for tetrahedral coordinated Fe³⁺; two transitions for octahedral coordinated Fe³⁺); however, all transitions could not be modelled in our XANES analyses because of low signal to noise intensity ratios for of most of these transitions; because of this we did not do this so as to not over interpret our data.

However, to indicate that not all electronic transitions allowed by the crystal-field theory are fittable in XANES, we added in lines 171-176 of the revised supplementary material:

"In fact, different coordination of FeII and FeIII should exhibit peak components for each electronic transition, according to the crystal-field theory (i.e., four transitions for FeII in square planar or tetrahedral coordination; three transitions octahedral coordinated FeII; two

transitions for tetrahedral coordinated FeIII; two transitions for octahedral coordinated FeIII); however, all transitions could not be modelled in XANES analyses because the signal to noise intensity ratio of most of these transitions was most often below the detection limits of the technique."

For all the above stated reasons we do not agree with reviewer 1 that the assigned Fe²⁺ and Fe³⁺ contributions to the FeS_{nano} phase are unconstrained by our XANES data. We have not made any major changes in the main text due to this reason but are happy to expand somewhat the explanation of how the fitting was done in the supplementary information file, if the editor finds this necessary.

R1 Comment 3: Maybe, the slight shift to higher E provides a qualitative hint.

R1 Comment 3 reply: Certainly, the shift to higher energies of the main peak and the increase in the edge jump indicates the presence of Fe³⁺ in the structure, as we have indeed explained in the previous version of the supplementary information which can now be found in lines 192-201 of the revised version:

"Furthermore, in supplementary figure 9b a shoulder feature less pronounced in FeSnano than in mackinawite spectrum could be observed. The higher energy of this feature in the spectrum for the FeSnano compared to that of mackinawite indicates a slightly higher proportion of FeIII present in FeSnano and this is in accordance also with our XPS data (Supplementary Table 6). Additional marked differences in the fingerprint of the near-edge feature were also observed between the FeSnano and mackinawite spectra, and the peak of the FeSnano phase was positioned at higher energy compared to that of mackinawite, again consistently with the above analysis. In summary, the fits for the XANES part of the FeSnano spectrum indicates the dominant presence of FeII in tetrahedral coordination and the presence of minor proportions of FeIII in octahedral environment."

Also this point was highlighted in the previous version of the supplementary Figure 9 caption, now in lines 474-479 of the revised version:

"the increase in the edge jump is shifted towards higher energies for FeSnano than for mackinawite, and this is also related to the higher proportion of FeIII present in FeSnano as also evidenced in our XPS data (Supplementary Table 6). In the near-edge region (c), there are clear differences in the shape of the main peak between FeSnano and mackinawite, and again the peak of the FeSnano phase is positioned at a higher energy, consistent with the presence of relatively more FeIII in its structure."

R1 Comment 4: The pre-edge is related to a 1s-3d transition and thus non-dipolar in nature. i.e. for a centrosymmetric site it is forbidden and only possible through quadrupolar mechanism (low intensity). It becomes dipole allowed by p-d mixing, e.g. by loss of centro-symmetry.

In essence, without presenting further sulfide model compounds and/or simulated spectra, the authors can only document a difference in site symmetry for Fe between FeS and mackinawite, with FeS having with a more centro-symmetric site than mackinawite. E.g. this could mean conversion from tetrahedral to octahedral symmetry or to square-planar symmetry. The authors need to rewrite this part and they should include more XANES data on sulfides at least from the literature (e.g. Prietzel et al. 2007 European Journal of Soil Science, 58, 1027).

R1 Comment 4 reply: In order to consider this reviewer’s 1 comment, we have attempted to compare our XANES data with the data of Prietzel et al 2007; however, all energies for sulfides in the Prietzel study are shifted towards higher values (~7113 eV, see table below) when compared to typical values reported for Fe²⁺ (~7112 eV). This could be because the analysed samples were natural materials that could have oxidized surfaces when exposed to air; Prietzel et al do not explain or confirm if their analyses were performed on fresh sample surfaces that reflect the natural reduced environment of formation, or if measures were taken to avoid surface oxidation prior to analyses. Nevertheless, they also analyzed two standard Fe²⁺ compounds (Fe^{II} lactate.H₂O and Fe^{II} oxalate.2H₂O) which certainly showed lower centroid pre-edge peak energies (~7112.6 eV) which were in agreement with the literature for Fe^{II} compounds. Although these latter standard compounds were useful for Fe²⁺ energies we have, contrary to the reviewer suggestion, not found that the sulfide reference samples presented by Prietzel et al 2007 were suitable for comparison with the sulfides from our work, as their results suggest either oxidation or influence by crystallization process that we do not have in our samples.

As the reviewer likely knows, in the iron sulfide system there are few mineral phases that contain Fe in both the 2+ and 3+ oxidation states and this makes a rigorous validation difficult; however, we have done the best we could with all available literature that fitted the description in terms of energy and sample handling and consider that the deconvolution approach we have applied based on literature suggestions is indeed valid to identify the contributions of Fe in the two oxidation states in the FeS_{nano} as discussed in *R1 Comment 2 reply*; we thus consider that in line with previous sulfide studies this approach demonstrated the difference in coordination in our new FeS_{nano} phase when compared to mackinawite.

Mineral/ sample	Fe Nominal Oxidation state	Centroid pre-edge peak, eV	Inflection point, eV	White line, eV
FeS _{nano}		7112.8	7117.6	7131.4
Mackinawite		7112.7	7120.6	7129.9
FeS (Prietzel et al 2007)	2+	7113.1	7116.6	7132.1
Pyrite (Prietzel et al 2007)	2+	7113.6	7117.9	7120.9

Pyrrhotite (Prietzal et al 2007)	2+	7113.3	7118.8	7120.0
Marcasite (Prietzal et al 2007)	2+	7113.4	7118.5	7121.4
Biotite (Prietzal et al 2007)	2+/3+	7113.8	7121.5	7122.9
			7125.0	7126.7
			7130.0	7131.0
Fe(II) lactate.H ₂ O (Prietzal et al 2007)	2+	7112.6	7123.0	7127.2
F(II) oxalate.2H ₂ O (Prietzal et al 2007)	2+	7112.5	7121.5	7127.2
K ₄ [Fe(CN) ₆].3H ₂ O (Westre et al 1997)	2+	7112.85		
Fe(prpep) ₂ .2CH ₃ OH (Westre et al 1997)	2+	7112.12		
FeSO ₄ .5H ₂ O (Westre et al 1997)	2+	7113.1	7120.5	7132.1
FeO (Wilke et al 2001)	2+	7112.45		
FeCO ₃ (Wilke et al 2001)	2+	7112.16		
FeSO ₄ .7H ₂ O (Wilke et al 2001)	2+	7112.16		

R1 Comment 5: Finally, the assignments to areas I, II, III with 1s-3d, 1s-4s, 1s-4p is not referenced at all, and only the first one is likely correct. 1s-4s transitions are also dipole-forbidden, so I doubt that it could be that strong. 1s-4p might be ok, but these states are already quite delocalized. In the absence of a DOS calculation, I would not quote any orbital assignment here. For the sake of the paper the terms I, II, III are fine. Please, do so.

R1 Comment 5 reply: We thank the reviewer for this comment, we deleted the assignment to the transitions and left the I, II, III notation. Please see lines 176-179 of the revised manuscript:

“Figure 3. XANES data of FeSnano. (a) Fe K-edge XANES spectra from FeSnano and mackinawite showing the (I) pre-edge, (II) increase in energy of the edge jump and (III) near-edge; (b) difference spectrum showing marked differences in the chemical environments between the two phases as revealed by the significant residual signal.”

We also have removed the text related to the transitions assigned to the areas I,II and III from the supplementary Figure 8, line 459-460 of the revised supplementary material:

“Supplementary Figure 8. Fe K-edge XANES spectra from FeSnano and mackinawite showing (I) a pre-edge, (II) an edge jump and (III) a near-edge feature. Furthermore, the Fe K-edge.....”

R1 Comment 6: The authors claim a difference in the Fe-O distance: 2.23 vs. 2.24 Å. Is this a significant difference given the limited EXAFS range for the FeS? I doubt that. Again, there is a difference to mackinawite, but only in the second shell. The mackinawite spectrum itself has no indication for signal by a second shell at all (Sup. Fig12 a +b), particularly not at 2.6 Å. In contrast, FeS shows some contribution by a second shell, as documented. So, I cannot follow the arguments of the authors from these data. There is no indication for Fe³⁺ in these EXAFS data, only some Fe-Fe contribution. Just from the Fourier Transform of the two spectra I would consider mackinawite to be more disordered than FeS, because it really shows no second-shell contribution. Please, rewrite accordingly or present stronger constraints.

R1 Comment 6 reply: We seemingly did not explain our approach clearly; we are not claiming a difference in the Fe-O distance: 2.23 vs. 2.24 Å (literature). We are actually comparing both Fe-S and Fe-Fe bond distances from the FeS_{nano} with those of mackinawite. Certainly, it could not have a difference between 2.23 and 2.24 Å within a +/- 0.01 Å resolution, but it could be if the error is towards a lower value and more if the reported Fe-S distances for mackinawite range between 2.24-2.26 Å. We are providing an explanation for a shorter bond scenario, to clarify this point we made changes to the text lines 165-174 in the revised manuscript:

“The local environment of Fe in our FeS_{nano} phase as analysed by extended X-Ray adsorption fine structure (EXAFS) revealed a first shell containing four S atoms in a tetrahedral coordination at a distance of 2.23 Å and a second shell containing at least two Fe atoms at 4.11 Å (Supplementary Fig. 12; Table 1 and Supplementary Table 11). There is a clear difference to mackinawite, in particular in the Fe-Fe bond distance (2.62 vs 4.11 Å) indicating a higher degree of disorder, likely as a consequence of the inclusion of Fe^{III} and polysulfides in its structure. The Fe-S bond distance derived in this study (2.23 Å) is marginally shorter than those reported in the literature (2.24 and 2.26 Å; 11), likely due to the change in oxidation state to accommodate tetrahedral and octahedral Fe (35-38; and supplementary text); however the resolution of the bond distance is +/- 0.01 Å.”

In relation to the reviewer's comment regarding the second shell, we realized that the Fourier transform of the supplementary figure 12 in the previous version was not clear enough; therefore, we have modified the supplementary figure 12 in the revised version of the supplementary material (line 497) to show the first and second shells for both FeS_{nano} and mackinawite data:

In addition, we want to point out to the reviewer that the second shell of EXAFS data for mackinawite is not particularly prominent as also shown by Lennie and Vaughan 1993 (See their data below, Fourier transform of the Fe K-edge EXAFS spectrum fitted with three shells; 2.24, 2.59 and 3.68 \AA).

Our fits are reliable as shown in the real part FT plot (line 497, supplementary figure 12c) and demonstrated by the statistics and the number of independent points in relation to that of the variables (please see also reply to comment 11).

R1 Comment 7: Line 95: Anaerobic XRD? Please, write “XRD patterns of FeS synthesized at anaerobic conditions”

R1 comment 7 reply: We thank the reviewer for pointing out this omission and are happy to comply with this change. We are also happy to explain and elaborate in detail in the text that we not only synthesized FeS under anaerobic conditions but that we also prepared, transported and analyzed each sample under strict anaerobic conditions using an airtight XRD holder.

This point was clarified in the revised manuscript text (lines 94-97):

"..... (a) XRD patterns of FeSnano synthesized under anaerobic conditions, the patterns show the three previously unknown low angle diffraction peaks; note absence of the characteristic Bragg peak for mackinawite at $\sim 5.0 \text{ \AA}$; the hump at $10.8 \text{ 2}\theta$ (8.2 \AA) suggests the presence of a agglomerated and poorly crystalline phase;...."

Also, the sample preparation description was expanded in lines 319-325:

"For XRD analysis, the as-synthesized anaerobic solid samples were filtered and re-dispersed in degassed ethanol and then mounted onto a flat silicon crystal inside an airtight Bruker XRD holder. This procedure was performed at all times in the anaerobic chamber with a strict and controlled oxygen free environment. The airtight holder was placed inside of an airtight container followed by a double sealed plastic bags to be transported to the XRD instrument for analysis maintaining each sample under stringent anaerobic conditions."

R1 Comment 8: Line 101: What is the Fe/S ratio based on? Simple intensity ratio or by using a matrix correction for analyzing the EDX pattern?

R1 comment 8 reply: Indeed, the Fe/S ratio is based on intensity ratio and we used no matrix correction, so the data was not biased.

R1 Comment 9: Figure 2b-g: What are the peak assignments based on? Are the peak positions constant no matter what the compound is? Especially, in case of the S- spectra the choice looks arbitrary to me with the limited info provided.

R1 comment 9 reply: This is a valid point and indeed in order to make adequate peak assignments we have carried out an extensive and thoughtful fitting process of all the $\text{Sp}_{3/2}$ spectra to model the components corresponding to monosulfides, disulfides and polysulfides.

For the Fe $2\text{p}_{3/2}$ spectra, peak components related to $\text{Fe}^{\text{II}}\text{-S}$ and $\text{Fe}^{\text{III}}\text{-S}$ species were considered. To build a model, several constrains were imposed (i.e., area of the peak, FWHM and peak position) and our fitting did not rely on the combination of individual spectra as a fingerprint but on the contribution of peaks with certain constrains that are characteristic for each specific species. The combination of components yielded a fit of the total envelope to the experimental data but did not relate to a specific model (for instance, mackinawite). Details of data collection and

processing were explained in in the previous versions of both, the manuscript and supplementary information but we can expand the fitting method description with the information below to make it clear.

The area under the curve (with a suitable baseline applied) was fitted with the minimum number of components to ensure a good fit. The components were constrained so that the FWHM was maintained, whilst allowing free fitting of peak area and position to ensure a good but scientifically viable fit. Where components were used to account for different bonding regimes, these were taken from literature values – however, this only introduce initial values of the peak positions and still allows for optimization of the peak areas in the fitting algorithm. All spectra were corrected to allow for any slight variations, using the C-C peak at 285 eV.

Therefore, to respond to the reviewer's comment, there was no an arbitrary assignment of the peak positions.

To clarify the XPS data processing further, we modified the text in the revised manuscript (lines 373-388):

“To build a model, several constrains were imposed (i.e., area of the peak, FWHM and peak position) and the fitting did not rely on the combination of individual spectra as a fingerprint but on the contribution of peaks with certain constrains that are characteristic for each specific species. The area under the curve (with a suitable baseline applied) was fitted with the minimum number of components to ensure best possible fit. The components were constrained so that the FWHM was maintained, whilst allowing free fitting of the peak area and position to ensure a good but scientifically viable fit. Where components were used to account for different bonding regimes, these were taken from literature – however, this approach only provided initial values for peak positions, yet still allowed for optimization of the peak areas through the fitting algorithm.

The full width at half-maximum (FWHM) was set at 1.8 eV for FeSnano and 1.0 eV for mackinawite. A satellite peak was included to fit the mackinawite spectrum. The S 2p spectra were fitted using the 2 p_{1/2} and 2 p_{3/2} doublets, the first one was set to be half the area of the second (Supplementary Table 6). The fits were compared with data from the literature for different iron sulfides (Supplementary Table 7, Supplementary Table 8). All spectra were corrected to allow for slight variations, using the C-C peak at 285 eV.”

R1 Comment 10: Line 363: Why only crystal-field splitting for Fe³⁺? Again, comparing the fit in the supplement on mackinawite with the one on FeS, gives the impression that the fit model are chosen arbitrarily, at least for a non-XPS reader. I would rather see a direct comparison of spectra including spectra of other model compounds, at least for other members of the FeS-family.

R1 comment 10 reply: As the reviewer likely knows, high spin Fe^{II} and Fe^{III} can be fitted with a multiplet structure derived from crystal field theory (Gupta and Sen,

1974, 1975); however, low spin Fe^{II} is unable to suffer multiple splitting and its contribution to the XPS spectrum is to one single peak as evidenced in multiple other studies (e.g., Nesbitt and Muir, 1994, Mycroft et al., 1990). The Fe^{II} of mackinawite has a low spin configuration and therefore, based on the fact that our FeS_{nano} is also a Fe-S phase precursor to mackinawite, we fitted Fe^{II} to only one peak. We agree with the reviewer that for a non-XPS expert, it would be useful to stress, on the XPS processing data section of the manuscript, that we employed previously used fitting methods as in the above-mentioned papers, and that we included only low spin Fe^{II}-S and Fe^{III}-S components in the fit. This approach yielded the best fit with the minimum number of components under the predetermined constraints (i.e., area of the peak, FWHM and peak position) as explained in our *R1 comment 9 reply below*.

To clarify this point we changed the text of the revised manuscript (line 371):

"...a single peak for low spin FeII species as in previous works (27, 47; Supplementary Table 6)."

R1 Comment 11: Line 403: why between 3 and 6 Å⁻¹. The EXAFS spectra are much longer. Is this a typo? If not the authors need to explain this in more detail.

R1 comment 11 reply: Certainly k-ranges can be longer, this is a valid point and we are happy to explain this further both here and in the revised supplementary material.

The fit of the mackinawite data was performed in the k-range between 3 and 10 Å⁻¹, while the fit for FeS_{nano} was performed in a smaller k-range (3 - 6.5 Å⁻¹) because the signal to noise above 6.5 Å⁻¹ was poor. We are aware that the k-range, in particular for the FeS_{nano} data is small but there is no restriction of using it for EXAFS fitting. Off course, there is a compromise in the number of variables that can be determined and the uncertainty of those determined increases. In the previous version of the manuscript, the number of independent points used for the fitting was twice the number of variables for the mackinawite data and 1.6 times for the FeS_{nano} data (supplementary Table 10 and Table 11); To test our approach we also performed, a second fit of FeS_{nano} to the greigite model in the k-range between 2.8 and 6.5 Å⁻¹; this was also achieved yielding twice the number of independent points with respect to the number of variables. This double fitting approach gave us reassurance that both our fit and the data were consistent.

Please note that with a k-range as small as 3.7 Å⁻¹ as for the FeS_{nano} data, we were able to resolve two different absorber - scatterer distances (Fe-S and Fe-Fe) bigger than 0.43 Å(Δr), which is far above the resolution given by the k-range used in accordance to the rule of thumb in which $K_{max} - k_{min} > \pi / (2 \times \Delta r)$.

For these reasons, the fits to the EXAFS data for FeS_{nano} and mackinawite are valid and correct, yet we revised the fit of FeS_{nano} data between 2.8 and 6.5 Å⁻¹ to ensure

that the independent points are twice the number of variables. The plots did not visually change but the fit results were updated in the Supplementary Table 11 (lines 367-370):

“Supplementary Table 11. EXAFS fitting parameters”

In addition, we have introduced a text to clarify this point in lines 259-264 in the revised supplementary material:

“It is important to note that although the fitting was performed over a small k-range (2.8 and 6.5 Å⁻¹ for FeSnano and 3-10 Å⁻¹ for mackinawite), the number of independent points used for the fitting was twice the number of variables (Supplementary Table 10 and Table 11). Within these k-ranges two different absorber - scatterer distances (Fe-S and Fe-Fe) bigger than 0.43 Å(Δr) were resolved, which is above the resolution given by the k-range used in accordance to the rule of thumb in which $K_{max} - k_{min} > \pi/(2x\Delta r)$.”

Reviewer #2 (Remarks to the Author):

As a generic reply to the comments of this reviewer, we highly appreciate the opinion of Martin Schoonen, and realize that his main concerns are centered around the following sentence (related to comments 1-3): “The entire interpretation depends on how well oxygen or other oxidants are avoided in the experimental setup and how well one can avoid O₂-driven alteration during any of the analyses”.

We show below in a point-by-point answer that we have at all stages taken extreme care (with many tests and careful measurements) that all experimental, sample handling, or sample analyses were done under anoxic settings. We may have not explained this well in the original manuscript as we assumed that the previous pedigree of the senior author with iron sulfide work may have been demonstrated that we can be fastidious but we have no problem explaining in minute detail (as we have done below) each and every step to document what we have done.

However, important to note that the assertion that we may have radicals in our water due to our MilliQ water preparation system is incorrect as in our system we do not have a UV lamp. Thus the whole argument is moot.

R2 Comment 1: The precipitation mechanism of Fe-S phases is important for a range of reasons as the authors indicate. The study uses a multi-modal analytical approach to determine the composition of the initial precipitate formed in the titration of a Fe(II) solution with an H₂S solution or a Fe(II) solution exposed to H₂S gas. The unexpected result is that a nano phase is formed which appears to contain both appreciable amounts of Fe(III) and polysulfides (i.e., S with an average oxidation state in excess of -II). This is unexpected because the phase this precursor leads to is mackinawite which is a slightly iron-rich FeS monosulfide phase, with all Fe(II) and S(-II). Hence, the results presented here indicate an initial precursor phase that is on balance more oxidized than the final product. The simplest interpretation of the data would be to assume that there is initially some minor amount of O₂ or other oxidant present that rapidly reacts and leads to the precursor, which then converts to mackinawite. The entire interpretation depends on how well oxygen or other oxidants are avoided in the experimental setup and how well one can avoid O₂-driven alteration during any of the analyses. The authors clearly realize this and in the supplemental material a considerable amount of attention is given to this issue.

However, this reviewer is concerned that the authors need to provide more details or present several control experiments to truly convince readers that the conclusions are based on experiments in which molecular oxygen or other oxidants did not play a role.

R2 Comment 1 reply: We thank Martin for these comments and indeed as he clearly states our “interpretation depends on how well oxygen or other oxidants are

avoided in the experimental setup and how well one can avoid O₂-driven alteration during any of the analyses”;

Precisely because we were aware of these issues, we have gone above and beyond normal precautions for avoiding any oxidation and are explaining below why the assertions that we must have had oxidation are not applicable to our experimental setup, sample handling or analyses. We give explanations to all raised points and can refute them with facts and thus consider that although we highly value Martin’s comments, they do not apply to our study.

R2 Comment 2: For example, it is posed in line 217 that deoxygenated water is used. There is NO indication how the water was deoxygenated, nor was a measurement of Dissolved Oxygen reported of this water. Placing DI with some slight amount of DO in an aerobic glove box does not ensure that the solution will be devoid of O₂. Sparging is probably the best method to remove DO, but earlier work by Butler et al shows that even with prolonged sparging some DO may remain, although the source of N₂ in this study may have a lower O₂ content because of the careful treatment with a trap. In any event, a control experiment to demonstrate the level of remaining O₂ is necessary. Furthermore, more recent work has shown that one has to also worry about UV irradiated water containing traces of H₂O₂. The study reports that Milli-Q water is used. Typically these units also include a UV lamp which leads to the formation of H₂O₂. Our experience is that one has to keep the water in the dark for a month or so to dissipate all H₂O₂ (Cohn, et al, Comparison of fluorescence-based techniques for the quantification of particle-induced hydroxyl radicals. *Particle and Fibre Toxicology*, 2008. 5.). It might be that the water used here was not irradiated, but those details should be spelled out.

R2 Comment 2 reply: Although we have not specified this in the submitted manuscript, we are happy to expand in a revised version. The preparation of O₂-free water was subject to a diligent and stringent process in all our experiments based on the myriad of studies before us that nicely documented how best to do this (including some of the papers that Martin cited). In our case, DI water was obtained from a Milli-Q Academic water system. This produces 18.2 MΩ water and the system in our laboratory is not equipped with an UV lamp as part of the deionization process, and thus the possible formation of H₂O₂ was not an issue for our experiments.

For each experiment, freshly drawn Milli-Q water was boiled under continuous stirring in specially designed scrubber flasks, while bubbling continuously with oxygen-free N₂ gas (grade 99,9995 that was in addition passed through a O₂ gas scrubbers). As an example, to produce 1L of water, we boiled it for 2 hours and then left it to cool down for another 2 hours, while continuously bubbling with the N₂ gas. Once at room temperature, the scrubber flask was sealed and the deoxygenated water was immediately transferred into the vacuum interlock of the Coy glove box.

There, 6 cycles of vacuum/N₂ injection were applied to remove any oxygen, prior to insertion of the slightly uncapped flask into the anaerobic chamber, in which the anaerobic environment was maintained with a N₂/H₂ mix (95:5 %) in a H₂ concentration of 3.8%. The O₂ free atmosphere was kept in the chamber through the standard Coy Lab catalyst, made of Al₂O₃ pellets coated with palladium. Any oxygen present in the chamber, readily reacts with the H₂ in the glove box to form water that adsorbs onto dried silica gel.

Inside the anaerobic chamber, an anaerobic gas monitor unit was placed equipped with a sound alarm when O₂ concentrations go above 1ppm. Throughout our experimental work, this monitor never showed any sign of oxygen in the chamber. In addition, prior to using the as above deoxygenated water in the experiments, we also removed an aliquot of the deoxygenated water and analyzed for dissolved oxygen concentrations using a Hack HQ30D portable DO meter and new factory calibrated luminescence / optical DO probe.

We have now included the procedure for preparation and transport of O₂- free water to the anaerobic chamber in lines 20-38 of the revised supplementary material, but we have refrained from doing so in the previous version of the manuscript as we were sure that this is a standard practice when working with iron sulfides. Lines 19-38:

“Preparation of O₂-free water

O₂-free water was prepared freshly for each experiment using 18.2 MΩ Milli-Q water from a Milli-Q Academic system. The water was boiled under continuous stirring in specially designed scrubber flasks, while bubbling continuously with O₂-free N₂ gas (grade 99,9995%). To produce 1L of water, we boiled it for 2 hours and then left it to cool down for another 2 hours, while continuously bubbling with the O₂-free N₂ gas. Once cooled to room temperature, the scrubber flask was sealed and the deoxygenated water was immediately transferred into the vacuum interlock of the glove box (CoyLaboratory Products Inc.). There, 6 cycles of vacuum/N₂ injection were applied to remove any oxygen, prior to insertion into the anaerobic chamber, in which the anaerobic environment was maintained with a N₂/H₂ mix (95:5 %) at a H₂ concentration of 3.8%. The O₂ free atmosphere was kept in the chamber through the standard Coy Lab catalyst, made of Al₂O₃ pellets coated with palladium. Any oxygen present in the chamber, readily reacted with the H₂ in the glove box to form water that adsorbs onto dried silica gel.

Inside the anaerobic chamber, an anaerobic gas monitor unit was placed, equipped with a sound alarm when O₂ concentrations raised above 1ppm. Throughout our experimental work, this monitor never showed any sign of oxygen in the chamber. In addition, prior to using the as above prepared deoxygenated water in an experiment, we tested an aliquot by analyzing its dissolved oxygen concentrations using a Hack HQ30D portable DO meter and a new factory calibrated luminescence / optical DO probe.”

R2 Comment 3: Perhaps a greater concern is possible alteration of samples during analysis. It is not clear how one would go about doing meaningful control experiments to reject the simplest interpretation that there is some initial oxidation. Other studies on this system have used in situ synchrotron-based pair distribution function analysis and concluded that 'amorphous FeS' had the mackinawite structure (Michel, F.M., et al., Short- to medium-range atomic order and crystallite size of the initial FeS precipitate from pair distribution function analysis. *Chemistry of Materials*, 2005. 17(25): p. 6246-6255). Because the authors are trying to evaluate a system with only 2% of the iron precipitated, the PDF technique would not be useful as an in situ technique as the signal is likely way too low. However, it might be useful to think how one might as much as possible conduct an in situ experiment. For example, would it be possible to use a Raman fiber optic probe to study the material in situ. Any evidence that would not rely on sample pretreatment followed by ex situ analysis would strongly lend support for their conclusion.

R2 comment 3 reply: As the reviewer indicated, the first thought of obtaining unexpected results is to believe that oxidation has occurred and reject results that do not follow the predictable; however, we can only state that exactly because of this issue we have taken utmost care in avoiding oxidation at each stage and in each one of the analyses, as we explained in detail (but maybe not detailed enough) in the previous version of the manuscript and supplementary material. We have implemented all possible checks and tests to demonstrate that no oxidation occurred and can only state that we are confident that no oxidation occurred.

We applied a meticulous procedure to make sure that oxygen from the atmosphere was not present at any time during synthesis, preparation, transport and analyses of samples. We can explain below each step in minute detail.

In particular, for the XRD and TEM analyses that were both carried out with samples transferred into special anaerobic cells inside the chamber (as explained in the supplementary material) we were pleased to see that both confirmed that a FeS_{nano} phase that formed at the beginning of the reaction process was not mackinawite. The same procedure (special anaerobic cell, transfer inside the chamber etc.) was used for Raman and XPS analyses. In addition, we monitored the Raman (Lines 421-434 in revised supplementary material), XPS (before and after etching) to document and demonstrate that there was no absorbed oxygen present; lines 114-128 in the revised manuscript, 116-129 in the revised supplementary material) and did the same tests (as explained in the SI) for the XPS analyses to confirm no beam induced changes. Finally, for the XAS analyses (detailed sample handling and analyses info in SI) we tested to document that no beam damage occurred and only after analyzing each individual spectrum, did we average and process all collected spectra

The reviewer appreciates the analytical limitations to analyze a tiny amount of forming solid in solution especially when the amount obtained is minute, and suggest that we should have attempted to carry out an in-situ characterizations. We thank the reviewer for the suggestion; however, we not only have actually tried to do this. We have not added the data in this manuscript because of the reasons below. We carried out synchrotron-based pair distribution function (PDF) analysis but the signal to noise ratios in the in situ samples (where the proportion of particles is minute compared to the solution signal) did not yield fittable results. The reviewer pointed out that Michel, et al 2005 used PDF to characterize amorphous FeS and concluded that it had a mackinawite structure. Our study is not comparable because the approach to synthesis was different, Mitchel et al. synthesized their FeS solids by acidifying a high pH (~9) HS⁻ solution with an Fe²⁺ solution to reach a minimal pH of 5.5. Performing the synthesis on that manner, undoubtedly leads to mackinawite formation (at pH 7). As the pH further decreases, the so formed solid mackinawite will likely start to dissolve yet it will maintain the same type of structure. As we explained, we started our work with a pH ~4.0 solution and slowly increased the pH observing the appearance of the FeS_{nano} at pH below 4.5. Then by increasing further the pH we obtained mackinawite. This was the way we could follow the formation of phases in the low pH regime prior to mackinawite and thus identify FeS_{nano}.

R2 Comment 4: Finally, I think it is important that the author present a more thorough analysis of the implications of the change in oxidation state of Fe and S in their system. If there is really nearly 40 % Fe(III), what are the energetics of this reverting back to Fe(II) to produce mackinawite.

R2 Comment 4 reply: We thank the reviewer for this consideration and this shows that we have not stated directly our reflection on this matter. In the lines 273-285 of previous version of the supplementary materials, we included a paragraph stating that for over 30 years, there has been considerable debate over Fe-S bond length and coordination in several Fe-S minerals (e.g., cubanite, greigite, chalcopyrite, pyrite and smythite). Mossbauer studies have shown that tetrahedral Fe in greigite can have a given oxidation state between 2+ and 3+; however, based on empirical calculations these oxidation states would yield longer Fe-S bond lengths than the actual 2.148 Å in greigite, and theoretically Fe^{IV} would yield a bond length of 2.144 Å (Hoggins and Steinfnik 1976; Shannon 1981) close to 2.148 Å as in greigite (Skinner, 1964). The increase in oxidation state is believed to occur as a result of electron back-donation from S atoms yielding shorter bond lengths (Rendon et al 1976). In our experiments, the presence of FeS_{nano} was always accompanied by the presence of disulfides and polysulfides, which allows the conversion from Fe^{II} to Fe^{III}. It is worth remembering that hydrogen sulfide is a very good oxidizing agent as it is able to accept electrons (Rickard 2015; In Pyrite a Natural History of Fool's Gold). In a system where reactions are not arrested at pH < 4.5, where FeS_{nano} remains stable, this phase would rapidly transform into mackinawite at room temperature with the

solely reducing environment imposed by the sulfide species; therefore the energetics required for the transformation from FeS_{nano} to mackinawite are not a limiting factor.

R2 Comment 5: With regard to the implications, the argument that this may be important on the early Earth is an overstatement. Even with a high CO₂ concentration in the atmosphere, the pH of the ocean would have been around 5 at best and the precipitation is likely to occur where highly alkaline solution flow into this ocean water. So the experiments here actually suggest that this precursor only plays a role below pH 4.1.

R2 Comment 5 reply: Indeed, we are happy to tone this implication down, as we are aware that it is always difficult to extrapolate highly controlled experiments to natural environment and even more so to environments on the early Earth, where we know little about the actual conditions where life emerged. It is however, well known that the formation of iron sulfides highly depend on the reaction rate between two species: Fe^{II}(aq) and aqueous sulfide species. We suggest that FeS_{nano} may be relevant in micro-niches like the highly reduced deep-ocean vent systems as possible transient phases that may have participated in life's catalyzed as at such vents today the pH is low and open structures like FeS_{nano} would facilitate an environment and highly reactive surface for the possible development of prebiotic molecules.

To clarify this point we have rewritten this reflection in lines 226-230 of the revised manuscript:

"The existence of metastable, transient low pH FeS_{nano} nanostructures may be relevant in micro-niches, for example in highly reduced deep-ocean hydrothermal vent systems"

R2 Comment 6: I think this is a well-conceived study, but given the implications more evidence that there was not some oxidant present either during the experiment or beam/irradiation induced alteration of the precursor is needed.

R2 Comment 5 reply: We hope that based on the above detailed steps in experiments, sample handling and sample analyses. In all our measurements, beam damage was avoided using the best available tests for each method (e.g., for XPS or Raman spectra, by collecting and comparing each individually collected scan before scans were summed up or averaged). For Raman and XPS, numerous tests and validation measurements were run before analyzing the final (publishable spectra), in order to quantitatively assess the best method and conditions for analysis of our samples. Nevertheless, the likely species to be found after an oxidation species would be Fe-O or sulfates and our spectra clearly document that these were absent at all times. Furthermore, we have also documented that any beam exposure did not induce transformation of the initial FeS_{nano} phase into ordered mackinawite.

Overall the authors have substantially improved the manuscript in regard of most comments. However, concerning the X-ray absorption spectroscopy part I still see considerable problems that are not acceptable and actually represent errors in the application of analytical procedures. I have listed my replies below.

If these comments are resolved the paper may well be accepted.

R1 comment 2:

Sorry, I was not clear enough with my comment: With “application to oxides” I meant compound where Fe is coordinated by oxygen. The ligand is important not the overall compound, glass melt etc. So, overall the reply is fine in this respect except for the following.

I do not agree with the answer on the pre-edge analysis: Sure, in previous work the pre-edge has been fitted with 2 or 3 peaks, but only to determine the intensity and centroid. The individual peaks had no real meaning in the analytical protocol. The Fe oxidation state was determined by the centroid and this was well constrained by many model compounds or external calibrations. The authors write themselves:

“To fit the XANES data, we used the method developed for oxides and later applied to silicate glasses, hydrous melts, sulfides and oxide-sulfide mixtures (9, 10, 14, 15). These studies have demonstrated that the centroid of the pre-edge (which is related to the positions of the individual components) is a reasonably accurate to estimate the FeII/FeIII ratio of an unknown sample composed of sulfides and oxides, when comparing the centroid position to those of the standards (15). Moreover, the peak positions of the deconvoluted peaks correlate with FeII and FeIII components, with the average position corresponding to the pre-edge position of the centroid (9, 10). **The approach of using the centroid of the peak has been well documented for sulfides, and the deconvolution of the peak has been demonstrated to be accurate to estimate the FeII and FeIII ratio.** Thus, we used this method to identify the presence and determine the proportions of these two oxidation states of Fe in the FeSnano.”

What is marked bold is not true: The centroid has not been used on sulfides. Prietzel et al. 2007 used it on sulfide ferrihydrite mixtures. “Deconvolution of the peak” was never used to determine Fe²⁺ and Fe³⁺, it was always the centroid used (determined by fitting 2-3 peaks). The authors actually report the centroid. Please, use it.

In the manuscript so far, the authors deconvolute the pre-edge with two peaks and use the intensity of the two peaks, one is interpreted for the ferric component the other for the ferrous one without any further calibration or constraint. They come up with 40% Fe³⁺/Fe_{tot} by using the intensity fractions of the two peaks shown in supplement figure 9. This approach has never been used and is complete nonsense in the way done. I have plotted the centroid values provided in supplementary table 9 on top of the variogram of Wilke et al. 2004. I had to shift the data by -0.7 eV due to systematic differences in energy calibration and background subtraction. The value is based on the centroid value of FeO used as a reference for Fe²⁺. From that graph it becomes clear that the FeSnano sample differs from mackinawite only by 10% in Fe³⁺/Fe_{tot} at most. This is also consistent with the statement by the authors that sulfides do not contain large amounts of Fe³⁺. The large difference between the two samples is the site symmetry, which likely goes from tetrahedral to octahedral (or square-planar?). The number of transitions predicted by multiplet cannot be fitted, however, there is severe difference in intensity between centro-symmetric and non-centrosymmetric sites. Thus, the difference between mackinawite and FeSnano is mostly related to this. With EXAFS the

authors may differentiate between octahedron and square planar by the number of fitted neighbours and the distance.

The manuscript should be rewritten accordingly.

Furthermore, the authors state that Fe²⁺ in mackinawite is in low spin state. In this case, the interpretation gets more complex. All previous authors except Westre et al. 1997. dealt high-spin Fe. The centroid method was never used on low spin Fe. The relation between centroid and intensity is likely very different from high spin Fe.

The consequences of the spin state are nicely illustrated in the following figures by Westre et al. 1997:

[Redacted]

[Redacted]

[Redacted]

[Redacted]

R1 comment 3:

The manuscript should be adopted to the previous comment

R1 comment 4:

Before comparing centroid values between different studies you should check the energy calibration. The values in the table are not comparable, because all studies used a different reference energy for the Fe-foil. The FeO sample in Wilke et al. 2001 was discussed by those authors to be partially oxidized, so don't compare to that, rather the other Fe²⁺ minerals with octahedral coordination.

R2 Comment 6 and EXAFS analysis:

During the fit of mackinawite the authors ended up with a S₀₂ value of 0.59 or 0.76. This an amplitude reduction factor that corrects for intrinsic amplitude losses. It has a 1:1 correlation with N, numbers of neighbors. As mackinawite is the reference sample with known coordination, it serves to determine S₀₂. Thus, this value should be used in later fits of unknown samples, right?

In the fit of the FeSnano phase the authors used S₀₂=1, instead of the value determined on the reference. What is the justification? Using a value of S₀₂=0.76 would yield N=6. The Fe-S distance might be a bit short for octahedral coordination, though.

Please, clarify.

Reviewer #2 (Remarks to the Author):

The authors have addressed my concerns expressed in my review. The rebuttal is presented in great detail and I have admit that the result remains surprising but the responses given by the authors have taken away my concern that the results are an artifact of the presence of O₂ in the system. The explanation of the electron back donation makes sense and it points to our somewhat simplistic assignments of oxidation states.

Overall the authors have substantially improved the manuscript in regard of most comments. However, concerning the X-ray absorption spectroscopy part I still see considerable problems that are not acceptable and actually represent errors in the application of analytical procedures. I have listed my replies below. If these comments are resolved the paper may well be accepted.

R1 comment 2:

Sorry, I was not clear enough with my comment: With “application to oxides” I meant compound where Fe is coordinated by oxygen. The ligand is important not the overall compound, glass melt etc. So, overall the reply is fine in this respect except for the following.

I do not agree with the answer on the pre-edge analysis: Sure, in previous work the pre-edge has been fitted with 2 or 3 peaks, but only to determine the intensity and centroid. The individual peaks had no real meaning in the analytical protocol. The Fe oxidation state was determined by the centroid and this was well constrained by many model compounds or external calibrations. The authors write themselves:

“To fit the XANES data, we used the method developed for oxides and later applied to silicate glasses, hydrous melts, sulfides and oxide-sulfide mixtures (9, 10, 14, 15). These studies have demonstrated that the centroid of the pre-edge (which is related to the positions of the individual components) is a reasonably accurate to estimate the FeII/FeIII ratio of an unknown sample composed of sulfides and oxides, when comparing the centroid position to those of the standards (15). Moreover, the peak positions of the deconvoluted peaks correlate with FeII and FeIII components, with the average position corresponding to the pre-edge position of the centroid (9, 10). **The approach of using the centroid of the peak has been well documented for sulfides, and the deconvolution of the peak has been demonstrated to be accurate to estimate the FeII and FeIII ratio.** Thus, we used this method to identify the presence and determine the proportions of these two oxidation states of Fe in the FeSnano.”

What is marked bold is not true: The centroid has not been used on sulfides. Prietzel et al. 2007 used it on sulfide ferrihydrite mixtures. “Deconvolution of the peak” was never used to determine Fe²⁺ and Fe³⁺, it was always the centroid used (determined by fitting 2-3 peaks). The authors actually report the centroid. Please, use it.

In the manuscript so far, the authors deconvolute the pre-edge with two peaks and use the intensity of the two peaks, one is interpreted for the ferric component the other for the ferrous one without any further calibration or constraint. They come up with 40% Fe³⁺/Fe_{tot} by using the intensity fractions of the two peaks shown in supplement figure 9. This approach has never been used and is complete nonsense in the way done. I have plotted the centroid values provided in supplementary table 9 on top of the variogram of Wilke et al. 2004. I had to shift the data by -0.7 eV due to systematic differences in energy calibration and background subtraction. The value is based on the centroid value of FeO used as a reference for Fe²⁺.

From that graph it becomes clear that the FeSnano sample differs from mackinawite only by 10% in Fe³⁺/Fe_{tot} at most. This is also consistent with the statement by the authors that sulfides do not contain large amounts of Fe³⁺. The large difference between the two samples is the site symmetry, which likely goes from tetrahedral to octahedral (or square-planar?).

The number of transitions predicted by multiplet cannot be fitted, however, there is severe difference in intensity between centro-symmetric and non-centrosymmetric sites. Thus, the difference between mackinawite and FeSnano is mostly related to this. With EXAFS the authors may differentiate between octahedron and square planar by the number of fitted neighbours and the distance.

The manuscript should be rewritten accordingly.

Furthermore, the authors state that Fe²⁺ in mackinawite is in low spin state. In this case, the interpretation gets more complex. All previous authors except Westre et al. 1997. dealt highspin Fe. The centroid method was never used on low spin Fe. The relation between centroid and intensity is likely very different from high spin Fe.

The consequences of the spin state are nicely illustrated in the following figures by Westre et al. 1997:

[Redacted]

R1 Comment 1 reply: First, we would like to thank to the Reviewer for the thoughtful comments that have considerably improved this manuscript. After further consideration, we have substantially reduced our XANES interpretation in view of the limited previous work on the pre-edge of Fe-S systems by removing the analysis approach that involved the deconvolution of the peaks. Thus, we now only use the centroid to derive the mean Fe oxidation state in FeS_{nano} but removed the text

in the manuscript and in the supplementary material related to the deconvolution. The table 1 (line 54) was modified regarding the XANES results. We have also clarified now in the supplementary material that this approach has been used only on oxides (line 152): “*For oxides, any variation in the oxidation state of Fe (Fe^{II} and Fe^{III}) could be identified because the two species have centroids at different energies that are normally separated by ~1.4eV (9).*”

R1 comment 3:

The manuscript should be adopted to the previous comment

R1 Comment 3 reply: Done, the manuscript now only discusses the XANES data employing the centroid approach.

R1 comment 4:

Before comparing centroid values between different studies you should check the energy calibration. The values in the table are not comparable, because all studies used a different reference energy for the Fe-foil. The FeO sample in Wilke et al. 2001 was discussed by those authors to be partially oxidized, so don't compare to that, rather the other Fe²⁺ minerals with octahedral coordination.

R1 Comment 4 reply: We used effectively the same energy calibration (Fe-Foil 7111.99 eV) as Wilke et al. (2001), Berry et al. (2003), Prietzel et al. (2007) and Humphreys et al. (2015), so our values should be comparable with theirs, i.e. approximately 0.9 eV higher than Wilke et al. (2001). To clarify this in the manuscript, we have added the following text in lines 400-401:

“Comparison with literature values revealed that our calibration was ~0.9 eV higher than Wilke et al 2001 (50).”

R2 Comment 6 and EXAFS analysis:

During the fit of mackinawite the authors ended up with a S02 value of 0.59 or 0.76. This an amplitude reduction factor that corrects for intrinsic amplitude losses. It has a 1:1 correlation with N, numbers of neighbors. As mackinawite is the reference sample with known coordination, it serves to determine S02. Thus, this value should be used in later fits of unknown samples, right?

In the fit of the FeSnano phase the authors used S02=1, instead of the value determined on the reference. What is the justification? Using a value of S02=0.76

would yield $N=6$. The Fe-S distance might be a bit short for octahedral coordination, though.

Please, clarify.

R2 Comment 6 reply: We agree that the S_{02} for the EXAFS would be expected to be similar for the mackinawite and FeS_{nano}, and thank the reviewer for bringing this to our attention. Indeed our initial fits did not yield good results using the same S_{02} for mackinawite (used as a standard) and the FeS_{nano}. The best fits for mackinawite always yielded low S_{02} (~ 0.6 or below) when fitting this parameter.

Although from our imaging results we were aware that both iron sulphide phases in our study (FeS_{nano} and mackinawite) were made up of particles in the nanoscale range we initially did not take this into consideration when fitting the EXAFS data. After discussion with Dr Fred Mosselmanns, and together with the reviewers comments we have re-fitted our spectra using an approach in which the coordination numbers are also a fitting parameters because the shells in our nanophases are not yet fully formed, and round coordination numbers are not expected in the fits.

When re-running the fits using this approach we obtained good results by using the same amplitude ($S_{02}=0.80$) for both the mackinawite and FeS_{nano} spectra. By refining the fits, we found reasonable values for all the variables including coordination numbers; however, some self-absorption could have affected the data and these number have to be taken with caution. It is worth mentioning however that the derived bond distances and ΔR values did not change when comparing with our previous fits.

We have amended the main text of the manuscript in several parts:

In Table 1 (line 54)

“Second shell FeII-Fe: 4.10, minimum 2 Fe atoms”

(lines 162-169)

“The local environment of Fe in our FeS_{nano} phase, as analysed by extended X-Ray absorption fine structure (EXAFS), revealed an average coordination of a first shell containing four S atoms in a tetrahedral coordination at a distance of 2.23 Å and a second shell containing at least two Fe atoms at 4.10 Å (Supplementary Fig. 12; Table 1 and Supplementary Table 11). There is a clear difference with mackinawite, in particular in the Fe-Fe bond distance (2.62 vs 4.10 Å), indicating a higher degree of disorder, likely as a consequence of the inclusion of FeIII and polysulfides in its structure. The Fe-S bond distance

derived in this study (2.23 Å) is similar to those reported in the literature (2.24 and 2.26 Å; 11)."

(lines 390-392)

"The particle size of the synthetic mackinawite used as a reference was ~ 5 nm as it was formed from the transformation of nanoparticulate FeS_{nano} and it was aged only for short periods of time."

(lines 408-416)

"In the fits, the amplitude parameter (S_0^2) was fixed to 0.80 for both paths, which was derived from the synthetic mackinawite standard. The energy shift (ΔE_0) was constrained to be the same for the two paths and it was set as a fitting parameter, as well as the disorder parameter in the distribution of inter-atomic distances (σ^2). Due to the small size of the particles, the coordination numbers (N) are not expected to be round figures and thus the coordination numbers could also be set as fitting parameters. To construct the fits, independent variables (N_{var}) were at least half of the independent points (N_{ind}). The data were then fitted through a least squares approach in the k -space between 2.8 and 6.5 Å⁻¹ using multiple k -weights. The quality of the fits....."

We have also modified the text in the supplementary material in the EXAFS result section

(lines 187-199) :

"The reduced $\chi^2= 7.04$ and R-factor=0.01 indicated a very good fit in agreement with the model. Furthermore, ΔR (<0.1Å), ΔE (<5 eV) and σ^2 were all within reasonable values. The interatomic distances for Fe-S of 2.23 Å and for Fe-Fe of 2.62 Å were also in a good agreement with the reported data (Fe-S = 2.24 Å and Fe-Fe = 2.63 Å or Fe-S = 2.23 Å and Fe-Fe = 2.60 Å (16, 17). Coordination numbers in the first shell (Fe-S) were 3.28 ± 0.12 and in the second shell 3.41 ± 0.65 ; however, these values must be used with caution as self-absorption problem in the data cannot be discarded given that the mackinawite spectrum had to be collected in fluorescence mode due to the small particle size or the freshly synthesized material. When this mackinawite EXAFS data was fitted to the greigite model the results were far less good as those obtained when fitting using the mackinawite model as the fit results for the second shell yielded extremely high σ^2 values. With these fits we could show that our experimental mackinawite data match well the mackinawite model from the literature."

(lines 210-213):

“Results of these comparisons are presented in Supplementary Fig. 12 and Supplementary Table 11. R-factors and χ^2 yielded comparable results for both models (R-factor=0.005, reduced $\chi^2= 178$ using the greigite model and R-factor=0.009 and reduced $\chi^2= 237$ using the mackinawite model).”

(lines 222-228):

“The fit parameters obtained using the greigite model were both reasonable at ΔR ($<0.1\text{\AA}$) and ΔE ($< \pm 5$ eV) and the value for σ^2 was similar ($\sim 0.006 \text{\AA}$) for the first and second shell. According to the greigite model, the first-shell has four S atoms around the central Fe atom, with interatomic distances of 2.23\AA and in a second shell, two Fe atoms with 4.10\AA bond distances. From the fit, coordination numbers (N) of 4.95 ± 0.19 for the first shell (Fe-S) and 2.09 ± 1.14 for the second shell (Fe-Fe) were derived. It is worth noting however that as mentioned before for the mackinawite fit, the coordination number....”

The fits in the Supplementary Tables 10 and 11 (lines 334 - 340), and in the Supplementary Figure 12 (lines 468-474) have been also replaced.

Reviewers' comments:

Reviewer #1 (Remarks to the Author):

XANES results:

Following previous comments, the XANES data are not used at all to quantitatively constrain the oxidation state of the FeSnano sample. Only qualitative arguments are now used to support the idea of a mixed Fe oxidation state, which is suggested from the XPS measurements.

The XANES analysis really lacks support by measurements on other model compounds. E.g. it would be very helpful to have a MEASURED spectrum of greigite, not just one taken from the literature. Mackinawite is listed as a compound with Fe²⁺ only, thus it is quite surprising to see also Fe³⁺ in that compound by XPS. This seems in line with earlier results (Boursiquot et al. 2001). Due to the small sampling depth of XPS it would be very important to get a second constraint on that. At the current stage the oxidation state is not well supported, in my view.

Thus, I strongly recommend changing the following lines:

Line 155: "This perfectly matched our Raman and XPS results (Fig. 2c; Supplementary Table 6), which also documented a mixed FeII/FeIII phase."

This is a useless statement in connection with the preceding statement on the pre-edge intensity and probably a remnant from the earlier version. Please, delete.

The authors should state: The difference in intensity is a clear indicator for different site symmetry. The small shift in centroid may indicate slightly higher Fe³⁺.

Supplement Line 162: "In both FeSnano and mackinawite, the centroid was positioned at slightly higher (7112.8 and 7112.7 eV) energies than typical for FeII (7112.1eV) suggesting the presence of at least some FeIII in both structures."

This is an unconstrained statement, the first two numbers are not comparable to the third.

The authors compare their centroid energy values to those typical in literature without correcting for bias. So, the typical literature value "7112.1" for Fe²⁺ converts to "7112.7" for their own FeO sample. I.e., if FeO is taken as reference for Fe²⁺, the FeSnano contains only 10-15% Fe³⁺. Another interpretation could be that mackinawite and FeSnano differ by 10-15% in Fe³⁺-content, if the authors do not trust their FeO sample. In the end, the authors cannot constrain the pre-edge position of an iron sulfide without any Fe³⁺ with the data that is presented.

Please, change the sentence to:

"Comparison of the centroid values of FeSnano and mackinawite (7112.8 vs 7112.7 eV) indicates a slightly higher proportion of Fe³⁺ for the FeSnano sample."

EXAFS results:

The authors have considerably modified the EXAFS analysis and present now a more differentiated view on the result. It seems obvious that the number of neighbors cannot be well fitted due to problems with the amplitude. This might well be related to the grain size of the material, which leads to a large contribution of Fe atoms at the surface.

Line 195-196: "This FeSnano phase consists of FeII in tetrahedral and FeIII in octahedral coordination,

a structure that is balanced by the presence of polysulfides besides the dominant monosulfides.”
Where is the evidence for Fe³⁺ in octahedral coordination? The EXAFS fit reveals a distance that is typical for tetrahedral coordination. Greigite has tetrahedral Fe²⁺ and octahedral Fe³⁺ (Skinner et al. 1964 and Li et al. 2014, Chemistry of Materials 26, (20) p5821-p5829). However, the authors show no evidence for Fe in an octahedral site (Fe-S at 2.42 and Fe-Fe at 3.5). So what is the difference between greigite and FeSnano? Apparently, the data of FeSnano do only show evidence for a tetrahedral site similar to the tetrahedral one of greigite. Please, delete any statement on octahedral Fe³⁺ (also in the abstract). It cannot be constrained by these data. It is a mere assumption based on the finding of Fe³⁺ by XPS.

The only evidence for octahedral symmetry, could come from the low pre-edge intensity, but that would need better constrains by measurements on model compounds like greigite or troilite etc. and/or simulation of spectra. At this stage the pre-edge intensity is not even considered as indicator. The authors present in the supplement a DFT-derived structural model, which basically not mentioned in the main text. With this structure model at hand, the authors could simulate the related XANES spectrum, easily, and compare it to other simulated spectra based on structural models of greigite, mackinawite, troilite etc.

In conclusion, all statements on Fe based on X-ray spectroscopy and XPS are rather weakly constrained by the data. Nonetheless, this study documents an important state for a nano-crystalline phase during the formation of sulfide crystals, already by the other techniques.

R1 Comment1:

XANES results:

Following previous comments, the XANES data are not used at all to quantitatively constrain the oxidation state of the FeSnano sample. Only qualitative arguments are now used to support the idea of a mixed Fe oxidation state, which is suggested from the XPS measurements.

The XANES analysis really lacks support by measurements on other model compounds. E.g. it would be very helpful to have a MEASURED spectrum of greigite, not just one taken from the literature.

Mackinawite is listed as a compound with Fe²⁺ only, thus it is quite surprising to see also Fe³⁺ in that compound by XPS. This seems in line with earlier results (Boursiquot et al. 2001). Due to the small sampling depth of XPS it would be very important to get a second constraint on that. At the current stage the oxidation state is not well supported, in my view.

R1 Comment 1 reply: We thank the Reviewer again for his further comments and appreciation of our work.

Given the method used to analyze the XANES data, we agree that it is not possible to quantify the Fe^{II}/Fe^{III} ratio in FeS_{nano}. However, the presence of a mixture of Fe^{II} and Fe^{III} can be derived from the centroid position, in comparison to that from Fe^{II} oxide systems, after calibration of the beamline using Fe foil. This qualitative assessment was quantitatively supported by XPS data, which is as mentioned indeed surface sensitive method to specifically quantify elemental speciation. Please see also our reply to R1 Comment 5.

We do not, however, agree with the Reviewer's suggestion that we should not compare with XANES data of greigite from the literature. Whereas the use of our own standards would be preferred, because their integrity can be assured, comparison with the literature is of course a generally accepted method that has been applied widely in the published literature (e.g., Hirst, C., Andersson, P.S., Shaw, S., Burke, I.T., Kutscher, L., Murphy, M.J., Maximov, T., Pokrovsky, O.S., Mörth, C.M. and Porcelli, D., 2017. Characterisation of Fe-bearing particles and colloids in the Lena River basin, NE Russia. *Geochimica et Cosmochimica Acta*, 213, pp.553-573).

Moreover, there is a large literature discussing and comparing Fe XANES data in different materials (e.g., Berry, et al 2003. XANES calibrations for the oxidation state of iron in a silicate glass. *American Mineralogist*, 88(7), pp.967-97; Zhang et al 2018.

Determination of Fe³⁺/ΣFe of XANES basaltic glass standards by Mössbauer spectroscopy and its application to the oxidation state of iron in MORB. Chemical Geology). We would therefore assert that, provided any calibration differences are accounted for, using literature standards is an effective and proven method.

Regarding the Reviewer's comment about mackinawite being listed as a compound with only Fe^{II} in the structure, we would like to emphasize that in this work the characterization analyses of mackinawite were performed from the separated solids at pH slightly higher than 5.5 and after the formation of FeS_{nano}. Given that this is an early formed nano-sized mackinawite obtained from FeS_{nano}, it is not surprising that some Fe^{III} is still present. The focus of this paper is not the structure of mackinawite but that of the new nanophase FeS_{nano}, yet we use mackinawite to show what this initial phase will transform to at higher pH values and longer times.

R1 Comment 2:

Thus, I strongly recommend changing the following lines:

Line 155: "This perfectly matched our Raman and XPS results (Fig. 2c; Supplementary Table 6), which also documented a mixed FeII/FeIII phase."

This is a useless statement in connection with the preceding statement on the pre-edge intensity and probably a remnant from the earlier version. Please, delete.

The authors should state: The difference in intensity is a clear indicator for different site symmetry. The small shift in centroid may indicate slightly higher Fe³⁺.

R1 Comment 2 reply:

We have changed the 'offending' sentence to (line 153-156): *The pre-edge in the FeS_{nano} spectrum was 65% less intense than in mackinawite (i.e., total integrated area in FeS_{nano}=0.09 and in mackinawite=0.26; Supplementary Table 9) indicating a different site symmetry. The shift in the centroid to higher energy indicates a slightly higher Fe^{III} content in the FeS_{nano}.*

R1 Comment 3:

Supplement Line 162: “In both FeSnano and mackinawite, the centroid was positioned at slightly higher (7112.8 and 7112.7 eV) energies than typical for FeII (7112.1eV) suggesting the presence of at least some FeIII in both structures.”

This is an unconstrained statement, the first two numbers are not comparable to the third.

The authors compare their centroid energy values to those typical in literature without correcting for bias. So, the typical literature value “7112.1” for Fe²⁺ converts to “7112.7” for their own FeO sample. I.e., if FeO is taken as reference for Fe²⁺, the FeSnano contains only 10-15% Fe³⁺. Another interpretation could be that mackinawite and FeSnano differ by 10-15% in Fe³⁺-content, if the authors do not trust their FeO sample. In the end, the authors cannot constrain the pre-edge position of an iron sulfide without any Fe³⁺ with the data that is presented.

Please, change the sentence to:

“Comparison of the centroid values of FeSnano and mackinawite (7112.8 vs 7112.7 eV) indicates a slightly higher proportion of Fe³⁺ for the FeSnano sample.”

R1 Comment 3 reply: We do not agree with the Reviewer’s assertion that our XANES data were not corrected for bias. All processing of XANES data were made after calibrating the energies using the Fe foil (7111.99 eV, i.e., the same calibration as Wilke et al. 2001) and therefore there is no bias in the analysis. We compared our data with the large database for Fe^{II} compounds provided by Wilke et al. 2001.

We have however, changed Supplement line 162 to: “In both FeS_{nano} and mackinawite, the centroid was positioned at slightly higher energies (7112.8 and 7112.7 eV) than is typical for Fe^{II} in oxides at 7112.1eV (9). This suggests the presence of at least some Fe^{III} in both structures, with a slightly higher content in FeS_{nano} (Supplementary Fig. 9a).”

R1 Comment 4:

EXAFS results:

The authors have considerably modified the EXAFS analysis and present now a more differentiated view on the result. It seems obvious that the number of neighbors cannot be well fitted due to problems with the amplitude. This might well be related to the grain size of the material, which leads to a large contribution of Fe atoms at the surface.

Line 195-196: "This FeSnano phase consists of FeII in tetrahedral and FeIII in octahedral coordination, a structure that is balanced by the presence of polysulfides besides the dominant monosulfides."

Where is the evidence for Fe³⁺ in octahedral coordination? The EXAFS fit reveals a distance that is typical for tetrahedral coordination. Greigite has tetrahedral Fe²⁺ and octahedral Fe³⁺ (Skinner et al. 1964 and Li et al. 2014, Chemistry of Materials 26, (20) p5821-p5829). However, the authors show no evidence for Fe in an octahedral site (Fe-S at 2.42 and Fe-Fe at 3.5). So what is the difference between greigite and FeSnano? Apparently, the data of FeSnano do only show evidence for a tetrahedral site similar to the tetrahedral one of greigite. Please, delete any statement on octahedral Fe³⁺ (also in the abstract). It cannot be constrained by these data. It is a mere assumption based on the finding of Fe³⁺ by XPS.

R1 Comment 4 reply: We have now deleted all text regarding octahedral Fe^{III} in both the manuscript and supplementary material. The difference between FeS_{nano} and other minerals, like mackinawite or greigite, was demonstrated by the diffraction data (i.e., XRD and electron diffraction).

R1 Comment 5:

The only evidence for octahedral symmetry, could come from the low pre-edge intensity, but that would need better constrains by measurements on model compounds like greigite or troilite etc. and/or simulation of spectra. At this stage the pre-edge intensity is not even considered as indicator.

The authors present in the supplement a DFT-derived structural model, which basically not mentioned in the main text. With this structure model at hand, the authors could simulate the related XANES spectrum, easily, and compare it to other simulated spectra based on structural models of greigite, mackinawite, troilite etc.

In conclusion, all statements on Fe based on X-ray spectroscopy and XPS are rather weakly constrained by the data. Nonetheless, this study documents an important state for a nano-crystalline phase during the formation of sulfide crystals, already by the other techniques.

R1 Comment 5 reply: We appreciate the positive comment from the Reviewer that highlights the importance of our work, revealing a new nano-crystalline precursor phase for sulfide minerals.

As the Reviewer suggested, we have modelled the XANES spectra for mackinawite, greigite, troilite, and the interactions of Fe-S layers with polysulfides (i.e. H₂S₂ and

H₂S₄), and show the results below in Figure 1 and 2 in this reply. In the supplementary information file Figure S8 and S9 we had shown how our data compares with mackinawite and greigite yet now below we also compared with other Fe-S phases. We have done this only in an attempt to simulate these interactions, yet, it is important to note that the models do neither fit nor represent a perfect match to the structure of the FeS_{nano} as described in this paper but are merely a means to illustrate how the different possible configurations could have affected the interpretation and fitting of our XANES spectra.

The pre-edge of the calculated XANES spectrum for mackinawite shows two peaks, instead of the single one observed in experimental data. As the models with intercalated polysulfides also consist of mackinawite-structured Fe-S layers, their simulated XANES spectra are therefore very close to that of mackinawite. However, they show wider peaks and the centroid position for both component peaks of the pre-edge is located at higher energies (0.1eV first peak, 0.4 eV second peak; please see Figure below). Similarly, the main peak of the polysulfide models is located at 1.9 eV higher than mackinawite, evidencing the effects of an open structure on the XANES spectrum (Figure 2 below). The pre-edges of greigite and troillite clearly exhibit only one peak, with centroids at 7112.8 and 7112.6, respectively. From these data, greigite has a pre-edge with the higher centroid energy value at 7112.8, with the main peak located at 7122.6 eV, which is 1.5 eV higher than modelled mackinawite and similar to the 1.6 eV difference between the main peak positions of FeS_{nano} and experimental mackinawite. As to the integrated intensity of the pre-edge, it is clear that greigite has the lowest intensity among all the models. However, the DFT calculations cannot discriminate between Fe^{II} and Fe^{III}, and we have therefore not included mixed Fe^{II}/Fe^{III} in the polysulfide models to observe the effect of the presence of Fe in both oxidation states on the pre-edge feature of the XANES spectrum.

Based on the above, the centroid of the pre-edge in the FeS_{nano} XANES spectrum agrees with the centroid of greigite, which does contain a mixture Fe^{II} and Fe^{III}, and is also likely to reflect the site symmetry of Fe in tetrahedral and octahedral coordination. However, the lack of reference materials to confirm Fe^{II} and Fe^{III} makes a thorough validation of the octahedral Fe in FeS_{nano} at this point impossible.

Figure 1 (for reply letter only). Pre-edge features of calculated XANES spectra and the centroid positions for Fe-S modelled structures obtained by DFT calculations.

Figure 2 (for reply letter only). XANES spectra for Fe-S modelled structures obtained by DFT calculations.

REVIEWERS' COMMENTS:

Reviewer #1 (Remarks to the Author):

Overall I am happy with the changes made. There is one final issue to be solved, then it's fine. Please, have a look into the attached file.